

# Semiconductor electron-phonon equations:
# A rung above Boltzmann in the many-body ladder

Gianluca Stefanucci⋆ and Enrico Perfetto

Dipartimento di Fisica, Università di Roma Tor Vergata,
Via della Ricerca Scientifica 1, 00133 Rome, Italy
INFN, Sezione di Roma Tor Vergata, Via della Ricerca Scientifica 1, 00133 Rome, Italy

⋆ gianluca.stefanucci@roma2.infn.it

## Abstract

Starting from the *ab initio* many-body theory of electrons and phonons, we go through a series of well defined simplifications to derive a set of coupled equations of motion for the electronic occupations and polarizations, nuclear displacements as well as phononic occupations and coherences. These are the semiconductor electron-phonon equations (SEPE), sharing the same scaling with system size and propagation time as the Boltzmann equations. At the core of the SEPE is the *mirrored* Generalized Kadanoff-Baym Ansatz (GKBA) for the Green's functions, an alternative to the standard GKBA which we show to lead to unstable equilibrium states. The SEPE treat coherent and incoherent degrees of freedom on equal footing, widen the scope of the semiconductor Bloch equations and Boltzmann equations, and reduce to them under additional simplifications. The new features of the SEPE pave the way for first-principles studies of phonon squeezed states and coherence effects in time-resolved absorption and diffraction experiments.



# 1   Introduction

In a recent work we have laid down the *ab initio* many-body theory of electrons and phonons [1], and derived the Kadanoff-Baym equations (KBE) [2,3] for the electronic and phononic Green's functions (GF). Solution of the KBE would provide us with a detailed understanding of the system's dynamics as we could extract electronic occupations and polarizations, nuclear displacements, phononic occupations and coherences as well as electronic and phononic spectral functions, life-times, quasi-particle renormalizations, satellites, etc. Unfortunately, the unfavourable scaling with the propagation time [4–10], which is at least cubic, continues to render the KBE challenging for *ab initio* simulations, though some progress has been made [11–13].

For crystals with a finite quasi-particle gap in the one-particle spectrum, like semiconductors or insulators, alternative (and approximate) theoretical frameworks to the KBE include the semiconductor Bloch equations (SBE) [14] and the Boltzmann equations (BE) [2,15]. At clamped nuclei both these frameworks can be derived from the KBE through well identifiable simplifications; in other words, we precisely understand what is missing. Orthodox derivations of the SBE are based on the cluster expansion [16], while the Wigner function [2,17] or the Fermi golden rule [18–20] are usually invoked to derive the BE. Alternatively one can use low-order diagrammatic expansions and the so called Generalized Kadanoff-Baym Ansatz (GKBA) for the electronic GF [21], which allows for mapping the KBE onto a single equation for the one-electron density matrix, see below. The situation is not as clear when electrons and phonons are treated on equal footing. The reason is twofold: Firstly the KBE for electrons and phonons have been only recently established [1]. Secondly, a GKBA for phonons has been proposed only a couple of years ago [22,23]. At present, both the SBE and BE for electrons *and* phonons must be regarded as semi-empirical frameworks since they are not derived from the *ab initio* Hamiltonian.

We here climb down the many-body ladder starting from the highest rung, i.e., the *ab initio* KBE. We use an alternative Ansatz to the GKBA, which we name the *mirrored* GKBA (MGKBA) [24]. The motivation for introducing the MGKBA arises from the observation that combining the GKBA with the Markov approximation results in unphysical outcomes and an

unstable equilibrium state. Through a series of well-defined simplifications we then derive a set of coupled equations for the electronic occupations and polarizations, nuclear displacements, and phononic occupations and coherences, which we name the *semiconductor electron-phonon equations* (SEPE). The SEPE scale like the SBE and BE with propagation time and system size. Their unique feature is a consistent treatment of phononic occupations and coherences as well as the inclusion of the renormalization of the electronic quasi-particle energies induced by the nuclear displacements. The former aspect opens the door to studies of phonon squeezed states in optically excited semiconductors [25–28] and time-dependent Debye-Waller factors [29–32]. The latter aspect is relevant for capturing the coherent modulation of time-resolved optical spectra of resonantly pumped semiconductors [33–37]. We finally elucidate the theoretical underpinnings of the SBE and BE, demonstrating how they emerge from the SEPE when additional simplifications are made.

The paper is organized as follows. In Sections 2 and 3 we briefly revisit the *ab initio* many-body theory of electrons and phonons [1]. In Section 4 we introduce the electronic and phononic density matrices, and derive their exact equations of motion in terms of electronic and phononic self-energies. The GKBA and MGKBA is discussed in Section 5 while self-energies and screening effects are presented in Section 6. Section 7 is the core of this work; it contains the derivation of the SEPE and a characterization of the solutions in the long-time limit. How to recover the SBE and BE is the topic of Section 8. A summary of the main findings and an outlook on future applications are drawn in Section 9.

## 2 Ab initio Hamiltonian for electrons and phonons

Let us consider a semiconductor or an insulator and assign a suitable basis to expand the electronic field operators and the nuclear displacement operators:

$$\hat{\psi}(\mathbf{x}) = \sum_{\mathbf{k}\mu} \Psi_{\mathbf{k}\mu}(\mathbf{x})\hat{d}_{\mathbf{k}\mu}, \tag{1}$$

$$\hat{U}_{\mathbf{n}s,j} = \frac{1}{\sqrt{M_s N}} \sum_{\mathbf{q}} e^{i\mathbf{q}\cdot\mathbf{n}} \sum_{\alpha} e_{s,j}^{\alpha}(\mathbf{q}) \, \hat{U}_{\mathbf{q}\alpha}. \tag{2}$$

In Eq. (1) the Bloch wavefunctions $\Psi_{\mathbf{k}\mu}(\mathbf{x}=\mathbf{r}\sigma)$ (position $\mathbf{r}$ and spin $\sigma$) are the eigenfunctions of a one-particle Hamiltonian with the periodicity of the crystal, e.g., the Kohn-Sham Hamiltonian (hence the index $\mu$ can be thought of as a band index). In Eq. (2) $\hat{U}_{\mathbf{n}s,j}$ is the displacement along direction $j = x, y, z$ of nucleus $s$ (with mass $M_s$) in cell $\mathbf{n}$, the total number of cells being $N$. The unit vectors $\mathbf{e}^{\alpha}(\mathbf{q})$, with components $e_{s,j}^{\alpha}(\mathbf{q})$, form an orthonormal basis for each $\mathbf{q}$. Although not necessary in this section, we can already choose these vectors to be the *normal modes* of the Born-Oppenheimer (BO) energy. The hermiticity of the operators $\hat{U}_{\mathbf{n}s,j}$ implies that $\hat{U}_{\mathbf{q}\alpha} = \hat{U}_{-\mathbf{q}\alpha}^{\dagger}$ and $\mathbf{e}^{\alpha*}(-\mathbf{q}) = \mathbf{e}^{\alpha}(\mathbf{q})$.

Let $\hat{P}_{\mathbf{q}\alpha} = \hat{P}_{-\mathbf{q}\alpha}^{\dagger}$ be the conjugate momentum of $\hat{U}_{\mathbf{q}\alpha}$, i.e., $[\hat{U}_{\mathbf{q}\alpha}, \hat{P}_{\mathbf{q}'\alpha'}^{\dagger}] = i\delta_{\mathbf{q},\mathbf{q}'}\delta_{\alpha\alpha'}$. The *ab initio* Hamiltonian for electrons and nuclei in the harmonic approximation can be written as (atomic units are used throughout this work) [1]

$$\hat{H} = \hat{H}_{0,e} + \hat{H}_{0,ph} + \hat{H}_{e-e} + \hat{H}_{e-ph}, \tag{3}$$

where

$$\hat{H}_{0,e} = \sum_{\mathbf{k}\mu\mu'} h_{\mu\mu'}(\mathbf{k})\hat{d}^{\dagger}_{\mathbf{k}\mu}\hat{d}_{\mathbf{k}\mu'}, \tag{4a}$$

$$\hat{H}_{0,ph} = \frac{1}{2}\sum_{\mathbf{q}\alpha\alpha'}\left(\hat{U}^{\dagger}_{\mathbf{q}\alpha}, \hat{P}^{\dagger}_{\mathbf{q}\alpha}\right)\begin{pmatrix} K_{\alpha\alpha'}(\mathbf{q}) & 0 \\ 0 & \delta_{\alpha\alpha'} \end{pmatrix}\begin{pmatrix} \hat{U}_{\mathbf{q}\alpha'} \\ \hat{P}_{\mathbf{q}\alpha'} \end{pmatrix} - \sum_{\mathbf{k}\mu\mu'}\sum_{\alpha}\rho^{\text{eq}}_{\mathbf{k}\mu'\mu}g_{\mathbf{0}\alpha,\mu\mu'}(\mathbf{k})\hat{U}_{\mathbf{0}\alpha}, \tag{4b}$$

$$\hat{H}_{e-e} = \frac{1}{2}\sum_{\substack{\mathbf{k}\mathbf{k}'\mathbf{q} \\ \mu\mu'\nu\nu'}} v_{\mathbf{k}+\mathbf{q}\mu\,\mathbf{k}'-\mathbf{q}\nu'\,\mathbf{k}'\nu\,\mathbf{k}\mu'}\hat{d}^{\dagger}_{\mathbf{k}+\mathbf{q}\mu}\hat{d}^{\dagger}_{\mathbf{k}'-\mathbf{q}\nu'}\hat{d}_{\mathbf{k}'\nu}\hat{d}_{\mathbf{k}\mu'}, \tag{4c}$$

$$\hat{H}_{e-ph} = \sum_{\mathbf{k}\mu\mu'}\sum_{\mathbf{q}\alpha}\hat{d}^{\dagger}_{\mathbf{k}\mu}\hat{d}_{\mathbf{k}-\mathbf{q}\mu'}g_{-\mathbf{q}\alpha,\mu\mu'}(\mathbf{k})\hat{U}_{\mathbf{q}\alpha}. \tag{4d}$$

In Eq. (4a), $h_{\mu\mu'}(\mathbf{k}) = \langle\mathbf{k}\mu|\frac{\hat{\mathbf{p}}^2}{2} + V(\hat{\mathbf{r}})|\mathbf{k}\mu'\rangle$ is the matrix element of the one-electron Hamiltonian, $V(\mathbf{r})$ being the potential generated by the nuclei in their equilibrium positions. Equation (4b) is the Hamiltonian of the bare phonons, with $K_{\alpha\alpha'}(\mathbf{q})$ the elastic tensor, $g_{\mathbf{0}\alpha,\mu\mu'}(\mathbf{k})$ the electron-phonon (*e-ph*) coupling at vanishing exchanged momentum (elastic scattering) and $\rho^{\text{eq}}_{\mathbf{k}\mu'\mu} = \langle\hat{d}^{\dagger}_{\mathbf{k}\mu}\hat{d}_{\mathbf{k}\mu'}\rangle$ the equilibrium one-electron density matrix. As pointed out in Ref. [1] the second term in Eq. (4b) plays a pivotal role in proving that the time-derivative of the nuclear momenta vanish in equilibrium. The explicit form of the elastic tensor is not important here, rather it is relevant to say that adding to it the equilibrium phononic self-energy $\Pi_{\mathbf{q}\alpha\alpha'}(\omega)$ in the clamped-nuclei approximation evaluated at $\omega = 0$ we have the exact identity (in the basis of the BO normal modes) [38, 39]

$$K_{\alpha\alpha'}(\mathbf{q}) + \Pi_{\mathbf{q}\alpha\alpha'}(\omega = 0) = \delta_{\alpha\alpha'}\omega^2_{\mathbf{q}\alpha}, \tag{5}$$

where $\omega^2_{\mathbf{q}\alpha}$ are the eigenvalues of the Hessian of the BO energy. The electron-electron (*e-e*) interaction is described by Eq. (4c), with

$$v_{\mathbf{k}+\mathbf{q}\mu\,\mathbf{k}'-\mathbf{q}\nu'\,\mathbf{k}'\nu\,\mathbf{k}\mu'} = \langle\mathbf{k}+\mathbf{q}\mu\,\mathbf{k}'-\mathbf{q}\nu'|v(\hat{\mathbf{r}},\hat{\mathbf{r}}')|\mathbf{k}'\nu\,\mathbf{k}\mu'\rangle, \tag{6}$$

the Coulomb scattering amplitudes. The *e-ph* interaction is accounted for by Eq. (4d), where the *e-ph* coupling is defined as

$$g_{-\mathbf{q}\alpha,\mu\mu'}(\mathbf{k}) = \langle\mathbf{k}\mu|\left.\frac{\partial V(\hat{\mathbf{r}})}{\partial U_{\mathbf{q}\alpha}}\right|_{\mathbf{U}=0}|\mathbf{k}-\mathbf{q}\mu'\rangle = g^*_{\mathbf{q}\alpha,\mu'\mu}(\mathbf{k}-\mathbf{q}). \tag{7}$$

We are interested in studying the dynamics of the system photoexcited by an external driving field

$$\hat{H}_{\text{drive}}(t) = \sum_{\mathbf{k}\mu\mu'}\Omega_{\mathbf{k}\mu\mu'}(t)\hat{d}^{\dagger}_{\mathbf{k}\mu}\hat{d}_{\mathbf{k}\mu'}, \tag{8}$$

with Rabi frequencies

$$\Omega_{\mathbf{k}\mu\mu'}(t) \equiv \frac{1}{2c}\langle\mathbf{k}\mu|\hat{\mathbf{p}}\cdot\mathbf{A}(\hat{\mathbf{r}},t) + \mathbf{A}(\hat{\mathbf{r}},t)\cdot\hat{\mathbf{p}} + \frac{1}{c}\mathbf{A}^2(\hat{\mathbf{r}},t)|\mathbf{k}\mu'\rangle. \tag{9}$$

We are here implicitly assuming that the driving field does not break the lattice periodicity. Commensurate distortions of the lattice can be studied using supercells and external fields having the periodicity of the supercell lattice.

# 3 Ab initio Kadanoff-Baym equations for electrons and phonons

We find it convenient to arrange the displacement and momentum operators into a two-dimensional vector

$$\hat{\boldsymbol{\phi}}_{\mathbf{q}\alpha} = \begin{pmatrix} \hat{\phi}^1_{\mathbf{q}\alpha} \\ \hat{\phi}^2_{\mathbf{q}\alpha} \end{pmatrix} = \begin{pmatrix} \hat{U}_{\mathbf{q}\alpha} \\ \hat{P}_{\mathbf{q}\alpha} \end{pmatrix}. \tag{10}$$

The *ab initio* KBE are coupled integro-differential equations for the electronic greater and lesser GFs

$$G^>_{\mathbf{k}\mu\mu'}(t,t') = -i\langle \hat{d}_{\mathbf{k}\mu,H}(t)\hat{d}^\dagger_{\mathbf{k}\mu',H}(t')\rangle, \tag{11a}$$

$$G^<_{\mathbf{k}\mu\mu'}(t,t') = i\langle \hat{d}^\dagger_{\mathbf{k}\mu',H}(t')\hat{d}_{\mathbf{k}\mu,H}(t)\rangle, \tag{11b}$$

and phononic greater and lesser GF ($i,j = 1,2$):

$$D^{ij,>}_{\mathbf{q}\alpha\alpha'}(t,t') = -i\langle \Delta\hat{\phi}^i_{\mathbf{q}\alpha,H}(t)\Delta\hat{\phi}^j_{-\mathbf{q}\alpha',H}(t')\rangle, \tag{12a}$$

$$D^{ij,<}_{\mathbf{q}\alpha\alpha'}(t,t') = -i\langle \Delta\hat{\phi}^j_{-\mathbf{q}\alpha',H}(t')\Delta\hat{\phi}^i_{\mathbf{q}\alpha,H}(t)\rangle, \tag{12b}$$

where $\Delta\hat{\phi}^j_{\mathbf{q}\alpha} = \hat{\phi}^j_{\mathbf{q}\alpha} - \langle\hat{\phi}^j_{\mathbf{q}\alpha}\rangle$ is the fluctuation operator. In these definitions the operators are in the Heisenberg picture with respect to the time-dependent Hamiltonian $\hat{H} + \hat{H}_{\text{drive}}(t)$. The KBE read (in matrix form) [1]

$$\left[ i\frac{d}{dt} - h(\mathbf{k},t) - \sum_\alpha g_{0\alpha}(\mathbf{k})U_{0\alpha}(t) \right] G^{\lessgtr}_{\mathbf{k}}(t,t') = \left[ \Sigma^{\text{R}}_{\mathbf{k}} \cdot G^{\lessgtr}_{\mathbf{k}} + \Sigma^{\lessgtr}_{\mathbf{k}} \cdot G^{\text{A}}_{\mathbf{k}} \right](t,t'), \tag{13a}$$

$$\left[ i\mathcal{J}\frac{d}{dt} - Q(\mathbf{q}) \right] D^{\lessgtr}_{\mathbf{q}}(t,t') = \left[ \Pi^{\text{R}}_{\mathbf{q}} \cdot D^{\lessgtr}_{\mathbf{q}} + \Pi^{\lessgtr}_{\mathbf{q}} \cdot D^{\text{A}}_{\mathbf{q}} \right](t,t'), \tag{13b}$$

where we use the symbol " · " to denote time-convolutions. In Eqs. (13)

$$h_{\mu\mu'}(\mathbf{k},t) = h_{\mu\mu'}(\mathbf{k}) + \Omega_{\mathbf{k}\mu\mu'}(t), \tag{14}$$

and

$$\mathcal{J}_{\alpha\alpha'} = \delta_{\alpha\alpha'}\begin{pmatrix} 0 & i \\ -i & 0 \end{pmatrix}, \qquad Q_{\alpha\alpha'}(\mathbf{q}) = \begin{pmatrix} K_{\alpha\alpha'}(\mathbf{q}) & 0 \\ 0 & \delta_{\alpha\alpha'} \end{pmatrix}. \tag{15}$$

The r.h.s. of the KBE contains the electronic self-energy $\Sigma_{\mathbf{k}}$ and the phononic self-energy $\Pi_{\mathbf{q}}$. As the electrons couple only to the nuclear displacements we have $\Pi^{ij}_{\mathbf{q}} = \delta_{i1}\delta_{j1}\Pi_{\mathbf{q}}$. Henceforth we use the same symbol $\Pi_{\mathbf{q}}$ to represent the $2 \times 2$ phononic self-energy and its (1,1) element; whether $\Pi_{\mathbf{q}}$ is a matrix or a scalar is evident from the context. The retarded (R) and advanced (A) correlators are defined in terms of the lesser and greater correlators according to $X^{\text{R/A}}(t,t') = \delta(t-t')X^\delta(t) \pm \theta(\pm t \mp t')[X^>(t,t') - X^<(t,t')]$, where $X^\delta$ is the weight of a possible singular part of $X$. For $G$ and $D$ the singular part is zero in all approximations. As we see later this is not the case for $\Sigma$ and $\Pi$. The KBE are coupled to the equation of motion of the nuclear displacement [see l.h.s. of Eq. (13a)] [1]

$$\frac{d^2}{dt^2}U_{0\alpha}(t) = -\sum_{\mathbf{k}\mu\nu} g_{0\alpha,\nu\mu}(\mathbf{k})\Delta\rho^<_{\mathbf{k}\mu\nu}(t) - \sum_{\alpha'} K_{\alpha\alpha'}(\mathbf{0})U_{0\alpha'}(t), \tag{16}$$

where $\Delta\rho^<_{\mathbf{k}\mu\nu}(t) = -iG^<_{\mathbf{k}\mu\nu}(t,t) - \rho^{\text{eq}}_{\mathbf{k}\mu\nu}$.

Solving the KBE with exact self-energies yield the exact two-times GFs. These provide information on the dynamics of carriers, phonon occupations and coherences as well as spectral properties relevant to time-resolved ARPES and Raman experiments. However, the time

non-locality of the self-energies represents a major numerical obstacle for a full two-times propagation. The time-convolutions in the r.h.s. of Eqs. (13) make any time-stepping algorithm scale at least cubically with the propagation time. In the following we introduce a series of simplifications leading to the SEPE, i.e., a couple system of ordinary differential equations for the electronic and phononic density matrices. The numerical solution of the SEPE scales linearly in time. The semiconductor Bloch equations and the Boltzmann equations follow from the SEPE by making additional simplifications.

# 4 Electronic and phononic density matrices

The electronic and phononic density matrices are proportional to the equal-time electronic and phononic GFs. We define them according to

$$\rho_{\mathbf{k}\mu\mu'}^{\gtrless}(t) \equiv -iG_{\mathbf{k}\mu\mu'}^{\gtrless}(t,t), \tag{17a}$$

$$\gamma_{\mathbf{q}\alpha\alpha'}^{\gtrless}(t) \equiv iD_{\mathbf{q}\alpha\alpha'}^{\gtrless}(t,t). \tag{17b}$$

Using the commutation rules for the electronic and nuclear operators we easily find

$$\rho_{\mathbf{k}\mu\mu'}^{>}(t) = \rho_{\mathbf{k}\mu\mu'}^{<}(t) - \delta_{\mu\mu'}, \tag{18a}$$

$$\gamma_{\mathbf{q}\alpha\alpha'}^{>}(t) = \gamma_{\mathbf{q}\alpha\alpha'}^{<}(t) + \mathcal{J}_{\alpha\alpha'}. \tag{18b}$$

The electronic density matrix $\rho_{\mathbf{k}\mu\mu'}^{<}(t) = \langle \hat{d}_{\mathbf{k}\mu',H}^{\dagger}(t)\hat{d}_{\mathbf{k}\mu,H}(t)\rangle$ is self-adjoint in the space of the band indices, i.e., $[\rho_{\mathbf{k}}^{<}]^{\dagger} = \rho_{\mathbf{k}}^{<}$. The diagonal entries are non-negative and determine the electronic occupations

$$f_{\mathbf{k}\mu}^{\text{el}}(t) \equiv \rho_{\mathbf{k}\mu\mu}^{<}(t), \tag{19}$$

while the off-diagonal entries provide information on the electronic polarization

$$p_{\mathbf{k}\mu\mu'}(t) \equiv \rho_{\mathbf{k}\mu\mu'}^{<}(t), \quad \mu \neq \mu'. \tag{20}$$

Similarly $[\rho_{\mathbf{k}}^{>}]^{\dagger} = \rho_{\mathbf{k}}^{>}$ has non-positive diagonal entries determining the negative of the hole occupations.

The phononic density matrix $\gamma_{\mathbf{q}\alpha\alpha'}^{ij,<}$ is self-adjoint in the direct-product space of the normal mode indices and components, i.e., $[\gamma_{\mathbf{q}}^{<}]^{\dagger} = \gamma_{\mathbf{q}}^{<}$, and similarly $[\gamma_{\mathbf{q}}^{>}]^{\dagger} = \gamma_{\mathbf{q}}^{>}$. To gain some more physical intuition on the phononic density matrix we write the displacements and momenta in terms of dressed phononic operators

$$\hat{U}_{\mathbf{q}\alpha} = \frac{1}{\sqrt{2\omega_{\mathbf{q}\alpha}}} \left( \hat{b}_{\mathbf{q}\alpha} + \hat{b}_{-\mathbf{q}\alpha}^{\dagger} \right), \tag{21a}$$

$$\hat{P}_{\mathbf{q}\alpha} = -i\sqrt{\frac{\omega_{\mathbf{q}\alpha}}{2}} \left( \hat{b}_{\mathbf{q}\alpha} - \hat{b}_{-\mathbf{q}\alpha}^{\dagger} \right). \tag{21b}$$

The commutation relations between $\hat{U}_{\mathbf{q}\alpha}$ and $\hat{P}_{\mathbf{q}'\alpha'}$ are satisfied *for any* $\omega_{\mathbf{q}\alpha} > 0$ provided that $[\hat{b}_{\mathbf{q}\alpha}, \hat{b}_{\mathbf{q}'\alpha'}^{\dagger}] = \delta_{\mathbf{q},\mathbf{q}'}\delta_{\alpha\alpha'}$. For later purposes we here take the $\omega_{\mathbf{q}\alpha}$'s to be the BO frequencies, see Eq. (5). The average values $U_{\mathbf{q}\alpha}(t) = \langle \hat{U}_{\mathbf{q}\alpha,H}(t)\rangle$ and $P_{\mathbf{q}\alpha}(t) = \langle \hat{P}_{\mathbf{q}\alpha,H}(t)\rangle$ are zero for all $\mathbf{q} \neq \mathbf{0}$ due to the fact that the lattice periodicity is preserved. Then, for all $\mathbf{q} \neq \mathbf{0}$ and for $\alpha = \alpha'$

we have

$$\gamma_{\mathbf{q}\alpha\alpha}^{11,<}(t) = \langle \Delta \hat{U}_{-\mathbf{q}\alpha,H}(t) \Delta \hat{U}_{\mathbf{q}\alpha,H}(t) \rangle$$

$$= \frac{1}{2\omega_{\mathbf{q}\alpha}} \left( f_{\mathbf{q}\alpha}^{\mathrm{ph}}(t) + f_{-\mathbf{q}\alpha}^{\mathrm{ph}}(t) + 1 + \Theta_{\mathbf{q}\alpha}(t) + \Theta_{\mathbf{q}\alpha}^{*}(t) \right), \qquad (22a)$$

$$\gamma_{\mathbf{q}\alpha\alpha}^{22,<}(t) = \langle \Delta \hat{P}_{-\mathbf{q}\alpha,H}(t) \Delta \hat{P}_{\mathbf{q}\alpha,H}(t) \rangle$$

$$= \frac{\omega_{\mathbf{q}\alpha}}{2} \left( f_{\mathbf{q}\alpha}^{\mathrm{ph}}(t) + f_{-\mathbf{q}\alpha}^{\mathrm{ph}}(t) + 1 - \Theta_{\mathbf{q}\alpha}(t) - \Theta_{\mathbf{q}\alpha}^{*}(t) \right), \qquad (22b)$$

$$\gamma_{\mathbf{q}\alpha\alpha}^{12,<}(t) = \langle \Delta \hat{P}_{-\mathbf{q}\alpha,H}(t) \Delta \hat{U}_{\mathbf{q}\alpha,H}(t) \rangle$$

$$= \frac{i}{2} \left( f_{\mathbf{q}\alpha}^{\mathrm{ph}}(t) - f_{-\mathbf{q}\alpha}^{\mathrm{ph}}(t) - 1 - \Theta_{\mathbf{q}\alpha}(t) + \Theta_{\mathbf{q}\alpha}^{*}(t) \right), \qquad (22c)$$

where we introduce the phononic occupations

$$f_{\mathbf{q}\alpha}^{\mathrm{ph}}(t) \equiv \langle b_{\mathbf{q}\alpha,H}^{\dagger}(t) b_{\mathbf{q}\alpha,H}(t) \rangle, \qquad (23)$$

and the *phononic coherences*

$$\Theta_{\mathbf{q}\alpha}(t) = \Theta_{-\mathbf{q}\alpha}(t) \equiv \langle b_{\mathbf{q}\alpha,H}(t) b_{-\mathbf{q}\alpha,H}(t) \rangle. \qquad (24)$$

An important property satisfied by the diagonal entries is

$$\gamma_{\mathbf{q}\alpha\alpha}^{ij,<}(t) = \gamma_{-\mathbf{q}\alpha\alpha}^{ji,>}(t). \qquad (25)$$

We use Eq. (25) in our subsequent derivations.

The exact equation of motion for $\rho_{\mathbf{k}}^{<}(t)$ can be derived by subtracting the lesser form of Eq. (13a) to its adjoint and then setting $t' = t$:

$$\frac{d}{dt}\rho_{\mathbf{k}}^{<}(t) + i\big[h_{\mathrm{qp}}(\mathbf{k},t), \rho_{\mathbf{k}}^{<}(t)\big] = -\int^{t} dt' \big[\Sigma_{\mathbf{k}}^{>}(t,t') G_{\mathbf{k}}^{<}(t',t) - \Sigma_{\mathbf{k}}^{<}(t,t') G_{\mathbf{k}}^{>}(t',t)\big] + \mathrm{h.c.}, \quad (26)$$

where

$$h_{\mathrm{qp}}(\mathbf{k},t) = h(\mathbf{k},t) + \sum_{\alpha} g_{0\alpha}(\mathbf{k}) U_{0\alpha}(t) + \Sigma_{\mathbf{k}}^{\delta}(t), \qquad (27)$$

is the so called quasi-particle Hamiltonian, and "h.c." stands for the hermitian conjugate. Often $\Sigma^{\delta}$ is evaluated in the Hartree plus statically screened exchange (HSEX) approximation. Similarly, the exact equation of motion for $\gamma_{\mathbf{q}}^{<}(t)$ can be derived by subtracting the lesser form of Eq. (13b) to its adjoint and then setting $t = t'$:

$$\frac{d}{dt}\gamma_{\mathbf{q}}^{<}(t) + i\big(\mathcal{J} Q_{\mathrm{qp}}(\mathbf{q},t)\gamma_{\mathbf{q}}^{<}(t) - \gamma_{\mathbf{q}}^{<}(t) Q_{\mathrm{qp}}(\mathbf{q},t)\mathcal{J}\big)$$

$$= \mathcal{J} \int^{t} dt' \big[\Pi_{\mathbf{q}}^{>}(t,t') D_{\mathbf{q}}^{<}(t',t) - \Pi_{\mathbf{q}}^{<}(t,t') D_{\mathbf{q}}^{>}(t',t)\big] + \mathrm{h.c.}, \qquad (28)$$

where

$$Q_{\mathrm{qp}}(\mathbf{q},t) \equiv Q(\mathbf{q}) + \Pi_{\mathbf{q}}^{\delta}(t), \qquad (29)$$

is the quasi-phonon, or dressed-phonon, Hamiltonian. A physically sensible approximation to $\Pi^{\delta}$ is discussed in Section 5.2.

# 5 Generalized Kadanoff-Baym Ansatz and its mirrored form

To close the equations of motion of the density matrices we have to transform the time off-diagonal $G^{\gtrless}$ and $D^{\gtrless}$ into functionals of $\rho^{<}$ and $\gamma^{<}$. In this way the r.h.s. in Eqs. (26) and (28) become functionals of $\rho^{<}$ and $\gamma^{<}$ since $\Sigma^{\gtrless}$ and $\Pi^{\gtrless}$ are functionals of $G^{\gtrless}$ and $D^{\gtrless}$.

## 5.1 Electrons

In the mid-1980's Lipavský et al. [21] proposed an Ansatz for the $G^{\gtrless}$-functional. In essence the idea is to manipulate and then modify the following exact relation for the noninteracting GF $G_0$ [3]:

$$G_{0,\mathbf{k}}^{<}(t,t') = G_{0,\mathbf{k}}^{R}(t,0)G_{0,\mathbf{k}}^{<}(0,0)G_{0,\mathbf{k}}^{A}(0,t'), \tag{30}$$

where

$$G_{0,\mathbf{k}}^{R}(t,t') = [G_{0,\mathbf{k}}^{A}(t',t)]^{\dagger} = -i\theta(t-t')T\{e^{-i\int_{t'}^{t}d\bar{t}\,h(\mathbf{k},\bar{t})}\}, \tag{31}$$

$T$ being the time-ordering operator. Using the group property $G_{0,\mathbf{k}}^{R}(t,0) = iG_{0,\mathbf{k}}^{R}(t,t')G_{0,\mathbf{k}}^{R}(t',0)$ for all $t > t' > 0$, and the like for the advanced GF, we find

$$G_{0,\mathbf{k}}^{<}(t,t') = -G_{0,\mathbf{k}}^{R}(t,t')\rho_{0,\mathbf{k}}^{<}(t') + \rho_{0,\mathbf{k}}^{<}(t)G_{0,\mathbf{k}}^{A}(t,t'), \tag{32}$$

where $\rho_{0,\mathbf{k}}^{<}(t) \equiv -iG_{0,\mathbf{k}}^{<}(t,t)$. An identical relation holds for $G_{0,\mathbf{k}}^{>}(t,t')$ provided that we replace $\rho_{0,\mathbf{k}}^{<}$ with $\rho_{0,\mathbf{k}}^{>}$. The *Generalized Kadanoff-Baym Ansatz* (GKBA) [21] amounts to approximate all interacting $G_{\mathbf{k}}^{\gtrless}$ in the collision integral of Eq. (26) (including those in the self-energy) as

$$G_{\mathbf{k}}^{\gtrless}(t,t') \simeq -G_{\mathbf{k}}^{R}(t,t')\rho_{\mathbf{k}}^{\gtrless}(t') + \rho_{\mathbf{k}}^{\gtrless}(t)G_{\mathbf{k}}^{A}(t,t'), \tag{33}$$

where [compare with Eq. (31)]

$$G_{\mathbf{k}}^{R}(t,t') = [G_{\mathbf{k}}^{A}(t',t)]^{\dagger} = -i\theta(t-t')T\{e^{-i\int_{t'}^{t}d\bar{t}\,h_{qp}(\mathbf{k},\bar{t})}\}. \tag{34}$$

This approximation to $G_{\mathbf{k}}^{R}$ corresponds to approximate $\Sigma_{\mathbf{k}}^{R}(t,t') \simeq \delta(t-t')\Sigma_{\mathbf{k}}^{\delta}(t)$. The GKBA is exact in the Hartree-Fock (HF) approximation and it is expected to be accurate when the average time between two consecutive collisions is longer than the quasi-particle decay time. For systems of only electrons the GKBA allows for closing the equation of motion Eq. (26) for any diagrammatic approximation to $\Sigma^{\gtrless}$. The GKBA equation of motion has been successfully applied in a large variety of physical situations. These include the nonequilibrium dynamics [40] and many-body localization [41] of Hubbard clusters, time-dependent quantum transport [42,43], equilibrium absorption of sodium clusters [44], real-time dynamics of the Auger decay [45], transient absorption [46–48] and carrier dynamics [49–52] of semiconductors, excitonic insulators out of equilibrium [53] as well as charge transfer [54] and charge migration [55–58] in molecular systems.

The GKBA is not the only Ansatz to transform $G^{\gtrless}$ into a functional of $\rho^{<}$. An equally simple and legitimate Ansatz can be obtained by observing that the group property of $G_{\mathbf{k}}^{R/A}$ in Eq. (34) implies

$$G_{\mathbf{k}}^{A}(0,t') = iG_{\mathbf{k}}^{A}(0,t)G_{\mathbf{k}}^{R}(t,t'), \qquad \forall\, t > t', \tag{35a}$$

$$G_{\mathbf{k}}^{R}(t,0) = -iG_{\mathbf{k}}^{A}(t,t')G_{\mathbf{k}}^{R}(t',0), \quad \forall\, t < t'. \tag{35b}$$

By the same arguments that lead to Eq. (33) we then obtain

$$G_{\mathbf{k}}^{\gtrless}(t,t') \simeq -\rho_{\mathbf{k}}^{\gtrless}(t)G_{\mathbf{k}}^{R}(t,t') + G_{\mathbf{k}}^{A}(t,t')\rho_{\mathbf{k}}^{\gtrless}(t'), \tag{36}$$

which we refer to as the *mirrored GKBA* (MGKBA). The GKBA and MGKBA can be obtained from one another by exchanging $G^{R} \leftrightarrow -G^{A}$. As pointed out in Ref. [24] any linear combination of the GKBA and MGKBA is also a legitimate Ansatz.

Like the GKBA also the MGKBA is exact at the HF level. In MGKBA the one-particle density matrix is on the left (right) of $G_{\mathbf{k}}^{R}$ ($G_{\mathbf{k}}^{A}$) and it is calculated at time $t$ ($t'$). Thus, the MGKBA equation of motion Eq. (26) can be written as

$$\frac{d}{dt}\rho_{\mathbf{k}}^{<}(t) + i\big[h_{qp}(\mathbf{k},t),\rho_{\mathbf{k}}^{<}(t)\big] = -\Gamma_{\mathbf{k}}^{el,>}(t)\rho_{\mathbf{k}}^{<}(t) - \Gamma_{\mathbf{k}}^{el,<}(t)\rho_{\mathbf{k}}^{>}(t) + \text{h.c.}, \tag{37}$$

where

$$\Gamma_{\mathbf{k}}^{\mathrm{el},\gtrless}(t) = \pm \int^t dt' \, \Sigma_{\mathbf{k}}^{\gtrless}(t,t') G_{\mathbf{k}}^{\mathrm{A}}(t',t), \tag{38}$$

can be interpreted as electronic scattering rates.

An important feature of both GKBA and MGKBA is that the exact relation

$$G_{\mathbf{k}}^{>}(t,t') - G_{\mathbf{k}}^{<}(t,t') = G_{\mathbf{k}}^{\mathrm{R}}(t,t') - G_{\mathbf{k}}^{\mathrm{A}}(t,t'), \tag{39}$$

is fulfilled independently of the choice of $G_{\mathbf{k}}^{\mathrm{R/A}}$; this is a direct consequence of Eq. (18a).

## 5.2 Phonons

The (M)GKBA for the electronic GF alone does not help in problems with electrons and phonons. The reason is that the electronic and phononic self-energies are functionals of $G$ and $D$ at different times. In order to close the equations of motion Eqs. (26) and (28) we need a (M)GKBA for phonons. This has been recently proposed by Karlsson et al. [22]. The idea is again to consider the noninteracting form of the phononic GF [1]

$$D_{0,\mathbf{q}}^{<}(t,t') = D_{0,\mathbf{q}}^{\mathrm{R}}(t,0)\mathcal{J}D_{0,\mathbf{q}}^{<}(0,0)\mathcal{J}D_{0,\mathbf{q}}^{\mathrm{A}}(0,t'), \tag{40}$$

and then use the group property of the noninteracting retarded/advanced GF

$$D_{0,\mathbf{q}}^{\mathrm{R}}(t,t') = [D_{0,\mathbf{q}}^{\mathrm{A}}(t',t)]^{\dagger} = -i\theta(t-t')\mathcal{J}\mathcal{W}_{0,\mathbf{q}}(t)\mathcal{W}_{0,\mathbf{q}}^{-1}(t'), \tag{41}$$

where $\mathcal{W}_{0,\mathbf{q}}(t) = T \exp\left[-i\int_0^t d\bar{t}\, Q(\mathbf{q})\mathcal{J}\right]$. Taking into account that $\mathcal{J}^2 = 1$, for any $t > t' > 0$ Eq. (41) implies

$$
\begin{aligned}
D_{0,\mathbf{q}}^{\mathrm{R}}(t,0) &= -i\mathcal{J}\mathcal{W}_{0,\mathbf{q}}(t) \\
&= -i\mathcal{J}\mathcal{W}_{0,\mathbf{q}}(t)\mathcal{W}_{0,\mathbf{q}}^{-1}(t')(i\mathcal{J})(-i\mathcal{J})\mathcal{W}_{0,\mathbf{q}}(t') \\
&= iD_{0,\mathbf{q}}^{\mathrm{R}}(t,t')\mathcal{J}D_{0,\mathbf{q}}^{\mathrm{R}}(t',0),
\end{aligned}
\tag{42}
$$

and the like for the advanced component. Following the same steps leading to the electronic GKBA in Eq. (32) we can rewrite Eq. (40) as

$$D_{0,\mathbf{q}}^{<}(t,t') = D_{0,\mathbf{q}}^{\mathrm{R}}(t,t')\mathcal{J}\gamma_{0,\mathbf{q}}^{<}(t') - \gamma_{0,\mathbf{q}}^{<}(t)\mathcal{J}D_{0,\mathbf{q}}^{\mathrm{A}}(t,t'), \tag{43}$$

where $\gamma_{0,\mathbf{q}}^{<}(t) = iD_{0,\mathbf{q}}^{<}(t,t)$. An identical relation holds for $D_{0,\mathbf{q}}^{>}(t,t')$ provided that we replace $\gamma_{0,\mathbf{q}}^{<}$ with $\gamma_{0,\mathbf{q}}^{>}$. The GKBA for phonons [22] amounts to approximate all interacting $D^{\gtrless}$ in the r.h.s. of Eq. (28) (including those in the self-energy) as

$$D_{\mathbf{q}}^{\gtrless}(t,t') \simeq D_{\mathbf{q}}^{\mathrm{R}}(t,t')\mathcal{J}\hat{\gamma}^{\gtrless}(t') - \hat{\gamma}^{\gtrless}(t)\mathcal{J}D_{\mathbf{q}}^{\mathrm{A}}(t,t'), \tag{44}$$

where [compare with Eq. (41)]

$$D_{\mathbf{q}}^{\mathrm{R}}(t,t') = [D_{\mathbf{q}}^{\mathrm{A}}(t',t)]^{\dagger} = -i\theta(t-t')\mathcal{J}\mathcal{W}_{\mathbf{q}}(t)\mathcal{W}_{\mathbf{q}}^{-1}(t'), \tag{45}$$

and $\mathcal{W}_{\mathbf{q}}(t) = T \exp\left[-i\int_0^t d\bar{t}\, Q_{\mathrm{qp}}(\mathbf{q},\bar{t})\mathcal{J}\right]$. This approximation to $D_{\mathbf{q}}^{\mathrm{R}}$ corresponds to approximate $\Pi_{\mathbf{q}}^{\mathrm{R}}(t,t') \simeq \delta(t-t')\Pi_{\mathbf{q}}^{\delta}(t)$.

Like in the electronic case the phononic GKBA is not the only Ansatz to transform $D^{\lessgtr}$ into a functional of $\gamma^{<}$. By definition

$$D_{\mathbf{q}}^{\mathrm{A}}(0,t') = D_{\mathbf{q}}^{\mathrm{A}}(0,t)(i\mathcal{J})D_{\mathbf{q}}^{\mathrm{R}}(t,t'), \qquad \forall\, t > t', \tag{46a}$$

$$D_{\mathbf{q}}^{\mathrm{R}}(t,0) = -D_{\mathbf{q}}^{\mathrm{A}}(t,t')(i\mathcal{J})D_{\mathbf{q}}^{\mathrm{R}}(t',0), \quad \forall\, t < t'. \tag{46b}$$

By the same arguments leading to Eq. (44) we then find an equally simple and legitimate Ansatz, which we refer to as the MGKBA for phonons:

$$D_{\mathbf{q}}^{\gtrless}(t,t') \simeq \gamma^{\gtrless}(t)\mathcal{J}D_{\mathbf{q}}^{\mathrm{R}}(t,t') - D_{\mathbf{q}}^{\mathrm{A}}(t,t')\mathcal{J}\gamma^{\gtrless}(t'). \tag{47}$$

In MGKBA the equation of motion Eq. (28) can be written as

$$\frac{d}{dt}\gamma_{\mathbf{q}}^{<}(t) + i\Big(\mathcal{J}Q_{\mathrm{qp}}(\mathbf{q},t)\gamma_{\mathbf{q}}^{<}(t) - \gamma_{\mathbf{q}}^{<}(t)Q_{\mathrm{qp}}(\mathbf{q},t)\mathcal{J}\Big) = -\Gamma_{\mathbf{q}}^{\mathrm{ph},>}(t)\gamma_{\mathbf{q}}^{<}(t) + \Gamma_{\mathbf{q}}^{\mathrm{ph},<}(t)\gamma_{\mathbf{q}}^{>}(t) + \mathrm{h.c.}, \tag{48}$$

where

$$\Gamma_{\mathbf{q}}^{\mathrm{ph},\gtrless}(t) = \mathcal{J}\int^{t} dt'\,\Pi_{\mathbf{q}}^{\gtrless}(t,t')D_{\mathbf{q}}^{\mathrm{A}}(t',t)\mathcal{J}, \tag{49}$$

can be interpreted as phononic scattering rates.

For phonons a physically sensible approximation to $\Pi^{\delta}$ is the clamped-nuclei plus static approximation [1], i.e.,

$$\Pi_{\mathbf{q}\alpha\alpha'}^{ij,\delta} = \delta_{i1}\delta_{j1}\sum_{\substack{\mathbf{k},\mathbf{k}' \\ \mu\mu' \\ \nu\nu'}} g_{\mathbf{q}\alpha,\mu\nu}(\mathbf{k})\chi_{\mathbf{k},\mathbf{k}',\substack{\nu\nu' \\ \mu\mu'}}^{\mathrm{R}}(\mathbf{q};\omega=0)g_{\mathbf{q}\alpha',\mu'\nu'}^{*}(\mathbf{k}'), \tag{50}$$

where $\chi$ is the response function at clamped nuclei (for the index structure see Fig. 1). Such self-energy renormalizes the block $(1,1)$ of the matrix $Q$, see Eq. (29), which in the basis of the BO normal modes becomes diagonal, see Eq. (5). Thus the whole matrix

$$Q_{\mathrm{qp},\alpha\alpha'}(\mathbf{q}) = \delta_{\alpha\alpha'}\begin{pmatrix} \omega_{\mathbf{q}\alpha}^{2} & 0 \\ 0 & 1 \end{pmatrix}, \tag{51}$$

becomes diagonal and the retarded GF simplifies to

$$D_{\mathbf{q}\alpha\alpha'}^{\mathrm{R}}(t,t') = i\delta_{\alpha\alpha'}\frac{\theta(t-t')}{2\omega_{\mathbf{q}\alpha}}\left[e^{i\omega_{\mathbf{q}\alpha}(t-t')}\begin{pmatrix} 1 & -i\omega_{\mathbf{q}\alpha} \\ i\omega_{\mathbf{q}\alpha} & \omega_{\mathbf{q}\alpha}^{2} \end{pmatrix} - e^{-i\omega_{\mathbf{q}\alpha}(t-t')}\begin{pmatrix} 1 & i\omega_{\mathbf{q}\alpha} \\ -i\omega_{\mathbf{q}\alpha} & \omega_{\mathbf{q}\alpha}^{2} \end{pmatrix}\right]. \tag{52}$$

This approximated form depends only on the time difference and can be Fourier transformed. It is easy to verify that $D_{\mathbf{q}\alpha}^{11,\mathrm{R}}(\omega) = 1/[(\omega+i\eta)^{2} - \omega_{\mathbf{q}\alpha}^{2}]$. We further note that also for phonons the exact property

$$D_{\mathbf{q}}^{>}(t,t') - D_{\mathbf{q}}^{<}(t,t') = D_{\mathbf{q}}^{\mathrm{R}}(t,t') - D_{\mathbf{q}}^{\mathrm{A}}(t,t'), \tag{53}$$

is fulfilled in both GKBA and MGKBA regardless of the choice of $D_{\mathbf{q}}^{\mathrm{R/A}}$; this is a direct consequence of Eq. (18b) and $\mathcal{J}^{2} = 1$.

The time-dependent renormalization of the phonon frequencies can be accounted for at least partially. As the response function is a functional of the electronic density matrix, the evaluation of $\chi$ at the time-dependent $\rho^{<}(t)$ yields a time-dependent $\Pi^{\delta}(t)$ and hence $\omega^{2}(t)$ through Eq. (5). In this work we discard such effect.

## 6 Electronic and phononic self-energies

Through the (M)GKBA for electrons and phonons the equations of motion (26) and (28) become integro-differential equations for $\rho^{<}$ and $\gamma^{<}$ for any diagrammatic approximation to the self-energies $\Sigma$ and $\Pi$. As a general remark we observe that the treatment of the Keldysh

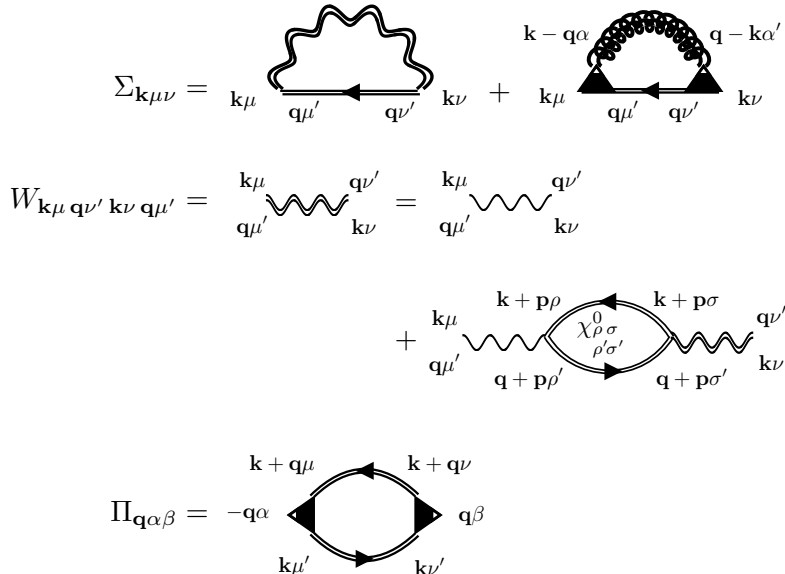

Figure 1: (Top) Electronic self-energy in the GW plus Fan-Migdal approximation. (Middle) RPA equation for the screened interaction $W$. (Bottom) Phononic self-energy. In the diagrams the oriented double line represents $G$, the double spring-line represents $D$, the wavy-line represents $v$, the double wavy-line represents $W$ and the triangle represents the $e\text{-}ph$ coupling.

components of the self-energies lacks consistency in (M)GKBA. The retarded/advanced self-energies are usually evaluated in some quasi-particle approximation (hence they are time-local) and have the only purpose of improving the retarded/advanced GFs in the Ansatzes. The lesser/greater self-energies do instead keep their diagrammatic form, see below.

We here consider the many-body approximation to $\Sigma$ and $\Pi$ which, as we see in Section 8, enables us to recover the Boltzmann equations. The $GW$ plus Fan-Migdal approximation for $\Sigma$ and the bare bubble approximation for $\Pi$ are illustrated in Fig. 1. In labelling the internal vertices we already take into account the conservation of the quasi-momentum.

For the $GW$ self-energy $\Sigma^{\mathrm{GW},\gtrless} = iG^{\gtrless}W^{\lessgtr}$ we need to express $W^{\lessgtr}$ in terms of $G^{<}$ and $G^{>}$. Starting from the Dyson equation $W = v + vPW$ one easily find $W^{\gtrless} = W^{\mathrm{R}}P^{\gtrless}W^{\mathrm{A}}$. To reduce the complexity of the equations we take $P = \chi^{0} = -iGG$ (RPA approximation) and evaluate $W^{\mathrm{R/A}}$ in the statically screened approximation. We have [14,59]

$$
\begin{aligned}
\Sigma^{\mathrm{GW},<}_{\mathbf{k}\mu\nu}(t,t') &= i \sum_{\mathbf{q}\nu'\mu'} W^{<}_{\mathbf{k}\mu\,\mathbf{q}\nu'\,\mathbf{k}\nu\,\mathbf{q}\mu'}(t,t') G^{<}_{\mathbf{q}\mu'\nu'}(t,t') \\
&= \sum_{\mathbf{q}\nu'\mu'} \sum_{\mathbf{p}\rho\sigma\rho'\sigma'} W_{\mathbf{k}\mu\,\mathbf{q}+\mathbf{p}\rho'\,\mathbf{k}+\mathbf{p}\rho\,\mathbf{q}\mu'} G^{>}_{\mathbf{q}+\mathbf{p}\sigma'\rho'}(t',t) G^{<}_{\mathbf{k}+\mathbf{p}\rho\sigma}(t,t') W_{\mathbf{k}+\mathbf{p}\sigma\,\mathbf{q}\nu'\,\mathbf{k}\nu\,\mathbf{q}+\mathbf{p}\sigma'} \\
&\quad \times G^{<}_{\mathbf{q}\mu'\nu'}(t,t').
\end{aligned}
\tag{54}
$$

The greater component of the self-energy is obtained by exchanging $> \leftrightarrow <$. We can add to the $GW$ self-energy the second-order exchange diagram with statically screened $W$ lines [60]. This amounts to replace $W_{\mathbf{k}_1\mu_1\,\mathbf{k}_2\mu_2\,\mathbf{k}_3\mu_3\,\mathbf{k}_4\mu_4} \to \frac{1}{2}[W_{\mathbf{k}_1\mu_1\,\mathbf{k}_2\mu_2\,\mathbf{k}_3\mu_3\,\mathbf{k}_4\mu_4} - W_{\mathbf{k}_1\mu_1\,\mathbf{k}_2\mu_2\,\mathbf{k}_4\mu_4\,\mathbf{k}_3\mu_3}]$ in Eq. (54).

Next we consider the Fan-Migdal self-energy in Fig. 1. We take a bare $e\text{-}ph$ coupling $g$ and later discuss how to dress it. We have

$$
\Sigma^{\mathrm{FM},<}_{\mathbf{k}\mu\nu}(t,t') = i \sum_{\mathbf{q}\nu'\mu'\alpha\alpha'} g_{\mathbf{q}-\mathbf{k}\alpha,\mu\mu'}(\mathbf{k}) D^{11,<}_{\mathbf{k}-\mathbf{q}\alpha\alpha'}(t,t') G^{<}_{\mathbf{q}\mu'\nu'}(t,t') g^{*}_{\mathbf{q}-\mathbf{k}\alpha',\nu\nu'}(\mathbf{k}),
\tag{55}
$$

where we use the property in Eq. (7). The Fan-Migdal self-energy calculated with a screened *e-ph* coupling $g^d = (1 + WP)g = (1 + v\chi)g$ is more involved. To account for screening effects to some degree we can make the clamped-nuclei plus static approximation used to estimate $\Pi^\delta$, i.e., $g^d = [1 + v\chi(\omega = 0)]g$. In this way the screened *e-ph* coupling is still time-local and the mathematical form of the Fan-Migdal self-energy is identical to Eq. (55) with $g \to g^d$. The greater component is obtained by exchanging $> \leftrightarrow <$.

We finally analyze the phononic self-energy. Keeping an eye on Fig. 1 we have

$$\Pi^{ii',<}_{\mathbf{q}\alpha\beta}(t,t') = -i\delta_{i1}\delta_{i'1} \sum_{\mathbf{k}\mu\nu\mu'\nu'} g_{\mathbf{q}\alpha,\mu'\mu}(\mathbf{k}) G^<_{\mathbf{q}+\mathbf{k}\mu\nu}(t,t') G^>_{\mathbf{k}\nu'\mu'}(t',t) g^*_{\mathbf{q}\beta,\nu'\nu}(\mathbf{k}), \qquad (56)$$

where we use Eq. (7). The greater component is obtained by exchanging $> \leftrightarrow <$. Dressing the *e-ph* coupling is here less straightforward since only one $g$ should be dressed [38,61–63]. Implementing the clamped-nuclei plus static approximation to only one $g$ would lead to the violation of the hermiticity properties of the self-energy. No violation occurs if the *nonlocal in time* dressed coupling is used. However, the equations of motion become more complex and, to the best of our knowledge, no efforts have been made to solve them thus far. Currently, all nonequilibrium state-of-the-art methods dress *both* $g$'s in Eq. (56), thereby suffering of a double counting problem. The SEPE do not resolve this issue either.

Through the GKBA for electrons and phonons the self-energies $\Sigma^{\text{GW},\lessgtr}(t,t')$, $\Sigma^{\text{FM},\lessgtr}(t,t')$ and $\Pi^{\lessgtr}(t,t')$ with $t > t'$ become functionals of $\rho^<(t')$ and $\gamma^<(t')$ with $t' < t$. Therefore the GKBA equations (26) and (28) carry memory, i.e., the density matrices at time $t$ depend on the history. This scheme has been recently implemented to study the relaxation of electrons and phonons in a photoexcited $MoS_2$ monolayer [64]. On the contrary the MGKBA leads to equations of motion with no memory since the self-energies become functionals of $\rho^<(t)$ and $\gamma^<(t)$.

# 7 Semiconductor electron-phonon equations

The SEPE are derived from the MGKBA equations after a series of simplifications, and they apply to crystals with a finite gap in the one-particle spectrum, like semiconductors or insulators. In Appendix B, we demonstrate that the *same* simplifications in the GKBA framework give rise to unphysical divergences and the inability to recover the Boltzmann equations. At least in this context, MGKBA is superior to GKBA.

The first simplification (S1) consists in using an MGKBA with diagonal density matrices and diagonal retarded/advanced GF – diagonality here refers to band indices for the electrons and mode indices for the phonons. For the electronic part we work in the eigenbasis of the equilibrium quasi-particle Hamiltonian, hence $h_{\text{qp},\mu\mu'}(\mathbf{k}) = \delta_{\mu\mu'}\epsilon_{\mathbf{k}\mu}$, and ignore the time-dependence of $h_{\text{qp}}(\mathbf{k},t)$ in Eq. (34). Then Eq. (36) becomes $G^{\lessgtr}_{\mathbf{k}\mu\nu}(t,t') = \delta_{\mu\nu}G^{\lessgtr}_{\mathbf{k}\mu\mu}(t,t')$, with

$$G^<_{\mathbf{k}\mu\mu}(t,t') = ie^{-i\epsilon_{\mathbf{k}\mu}(t-t')}\big[\theta(t-t')f^{\text{el}}_{\mathbf{k}\mu}(t) + \theta(t'-t)f^{\text{el}}_{\mathbf{k}\mu}(t')\big]. \qquad (57)$$

The expression for $G^>_{\mathbf{k}\mu\nu}(t,t')$ is identical provided that $f^{\text{el}}_{\mathbf{k}\mu} \to f^{\text{el}}_{\mathbf{k}\mu} - 1$, see Eq. (18a). For the phononic part we work in the basis of the BO normal modes. We observe that the off-diagonal elements of the phononic density matrix play a role in the field of thermal transport [65, 66], which is not the focus of the present work. Taking into account Eqs. (22) and (52), the

phononic MGKBA in Eq. (47) yields $D^{\lessgtr}_{\mathbf{q}\alpha\beta}(t,t') = \delta_{\alpha\beta} D^{\lessgtr}_{\mathbf{q}\alpha\alpha}(t,t')$, with

$$D^{11,\lessgtr}_{\mathbf{q}\alpha\alpha}(t,t') = \frac{\theta(t-t')}{2i\omega_{\mathbf{q}\alpha}}\left[B^{\lessgtr*}_{\mathbf{q}\alpha}(t)e^{-i\omega_{\mathbf{q}\alpha}(t-t')} + B^{\gtrless}_{-\mathbf{q}\alpha}(t)e^{i\omega_{\mathbf{q}\alpha}(t-t')}\right]$$
$$+ \frac{\theta(t'-t)}{2i\omega_{\mathbf{q}\alpha}}\left[B^{\lessgtr}_{\mathbf{q}\alpha}(t')e^{-i\omega_{\mathbf{q}\alpha}(t-t')} + B^{\gtrless*}_{-\mathbf{q}\alpha}(t')e^{i\omega_{\mathbf{q}\alpha}(t-t')}\right], \tag{58}$$

$$D^{12,\lessgtr}_{\mathbf{q}\alpha\alpha}(t,t') = \frac{\theta(t-t')}{2}\left[B^{\lessgtr*}_{\mathbf{q}\alpha}(t)e^{-i\omega_{\mathbf{q}\alpha}(t-t')} - B^{\gtrless}_{-\mathbf{q}\alpha}(t)e^{i\omega_{\mathbf{q}\alpha}(t-t')}\right]$$
$$+ \frac{\theta(t'-t)}{2}\left[C^{\lessgtr}_{\mathbf{q}\alpha}(t')e^{-i\omega_{\mathbf{q}\alpha}(t-t')} - C^{\gtrless*}_{-\mathbf{q}\alpha}(t')e^{i\omega_{\mathbf{q}\alpha}(t-t')}\right], \tag{59}$$

and

$$B^{<}_{\mathbf{q}\alpha}(t) = f^{\text{ph}}_{\mathbf{q}\alpha}(t) + \Theta_{\mathbf{q}\alpha}(t), \tag{60a}$$

$$B^{>}_{\mathbf{q}\alpha}(t) = f^{\text{ph}}_{\mathbf{q}\alpha}(t) + 1 + \Theta_{\mathbf{q}\alpha}(t), \tag{60b}$$

$$C^{<}_{\mathbf{q}\alpha}(t) = f^{\text{ph}}_{\mathbf{q}\alpha}(t) - \Theta_{\mathbf{q}\alpha}(t), \tag{60c}$$

$$C^{>}_{\mathbf{q}\alpha}(t) = f^{\text{ph}}_{\mathbf{q}\alpha}(t) + 1 - \Theta_{\mathbf{q}\alpha}(t). \tag{60d}$$

As we see later, the GF $D^{22}$ is not needed.

In the following we derive the SEPE for the electronic occupations and polarizations, nuclear displacements, and phononic occupations and coherences. Without any loss of generality, we assume that the system is initially in equilibrium at negative times and it is subsequently perturbed by a driving field of finite duration $T_{\text{drive}}$; hence the vector potential $\mathbf{A}(\mathbf{r},t) = 0$ for $t < 0$ and for $t > T_{\text{drive}}$.

## 7.1 Electronic occupations and polarizations

We begin with by decomposing $h + \Sigma^{\delta}$ into its diagonal and off-diagonal components

$$h_{\mu\nu}(\mathbf{k}) + \Sigma^{\delta}_{\mathbf{k}\mu\nu}(t) = \delta_{\mu\nu}\tilde{\epsilon}_{\mathbf{k}\mu}(t) + (1-\delta_{\mu\nu})\Delta\Sigma^{\delta}_{\mathbf{k}\mu\nu}(t), \tag{61}$$

where

$$\tilde{\epsilon}_{\mathbf{k}\mu}(t) = \epsilon_{\mathbf{k}\mu} + \Delta\Sigma^{\delta}_{\mathbf{k}\mu\mu}(t), \tag{62}$$

and

$$\Delta\Sigma^{\delta}_{\mathbf{k}\mu\nu}(t) = \sum_{\mathbf{k}'\mu'\nu'}\left(v_{\mathbf{k}\mu\mathbf{k}'\mu'\mathbf{k}'\nu'\mathbf{k}\nu} - W_{\mathbf{k}\mu\mathbf{k}'\mu'\mathbf{k}\nu\mathbf{k}'\nu'}\right)\left[\rho^{<}_{\mathbf{k}'\nu'\mu'}(t) - \rho^{<}_{\mathbf{k}'\nu'\mu'}(0)\right]. \tag{63}$$

Equation (63) is the change of the HSEX potential. At time $t = 0$ the Rabi frequencies and the nuclear displacements vanish, and therefore, see Eqs. (14) and (27),

$$h_{\text{qp},\mu\mu'}(\mathbf{k},0) = h_{\mu\mu'}(\mathbf{k}) + \Sigma^{\delta}_{\mathbf{k}\mu\mu'}(0) = \delta_{\mu\mu'}\epsilon_{\mathbf{k}\mu}, \tag{64}$$

as it should.

Let $S^{\text{el}}_{\mathbf{k}}(t)$ be the r.h.s. of Eq. (26); we call this quantity the *electronic scattering term*. We then have

$$\frac{d}{dt}f^{\text{el}}_{\mathbf{k}\mu} + i\sum_{\nu\neq\mu}\left(\Omega^{\text{ren}}_{\mathbf{k}\mu\nu}p_{\mathbf{k}\nu\mu} - p_{\mathbf{k}\mu\nu}\Omega^{\text{ren}}_{\mathbf{k}\nu\mu}\right) = S^{\text{el}}_{\mathbf{k}\mu\mu}, \tag{65}$$

$$\frac{d}{dt}p_{\mathbf{k}\mu\nu} + i\left(\tilde{\epsilon}_{\mathbf{k}\mu} - \tilde{\epsilon}_{\mathbf{k}\nu}\right)p_{\mathbf{k}\mu\nu} + i\Omega^{\text{ren}}_{\mathbf{k}\mu\nu}\left(f^{\text{el}}_{\mathbf{k}\nu} - f^{\text{el}}_{\mathbf{k}\mu}\right) + i\sum_{\nu'\neq\nu}\Omega^{\text{ren}}_{\mathbf{k}\mu\nu'}p_{\mathbf{k}\nu'\nu} - i\sum_{\nu'\neq\mu}p_{\mathbf{k}\mu\nu'}\Omega^{\text{ren}}_{\mathbf{k}\nu'\nu} = S^{\text{el}}_{\mathbf{k}\mu\nu}, \tag{66}$$

where we define the renormalized Rabi frequencies

$$\Omega_{\mathbf{k}\mu\nu}^{\mathrm{ren}}(t) \equiv \Omega_{\mathbf{k}\mu\nu}(t) + \Delta\Sigma_{\mathbf{k}\mu\nu}^{\delta}(t) + \sum_{\alpha} g_{0\alpha,\mu\nu}(\mathbf{k})U_{0\alpha}(t). \tag{67}$$

The equations of motion Eqs. (65) and (66) with $S^{\mathrm{el}} = 0$ are equivalent to solving the time-dependent HSEX equations, leading to the Bethe-Salpeter equation in the linear response regime [3, 67]; they have been recently implemented to investigate the dynamics of coherent excitons [68–70]. The diagonal part of the scattering term is the sum of the *GW* and Fan-Migdal contributions:

$$S_{\mathbf{k}\mu\mu}^{\mathrm{el}}(t) = S_{\mathbf{k}\mu\mu}^{\mathrm{GW}}(t) + S_{\mathbf{k}\mu\mu}^{\mathrm{FM}}(t). \tag{68}$$

By using the simplified MGKBA of Eqs. (57) and (58) when calculating the self-energies in Eqs. (54) and (55), both contributions in Eq. (68) can be expressed in terms of $f^{\mathrm{el}}$, $f^{\mathrm{ph}}$ and $\Theta$. The second simplification (S2) of the SEPE is the Markov approximation for all exponential integrals, i.e., $\int_0^t e^{iEt} \simeq \pi\delta(E)$. After some straightforward algebra we find

$$S_{\mathbf{k}\mu\mu}^{\mathrm{GW}} = 2\pi \sum_{\mathbf{q}\mu'} \sum_{\mathbf{p}\nu\nu'} \left| W_{\mathbf{k}\mu\,\mathbf{q}+\mathbf{p}\,\nu'\,\mathbf{k}+\mathbf{p}\,\nu\,\mathbf{q}\mu'} \right|^2 \delta\big(\epsilon_{\mathbf{k}+\mathbf{p}\,\nu} + \epsilon_{\mathbf{q}\mu'} - \epsilon_{\mathbf{q}+\mathbf{p}\,\nu'} - \epsilon_{\mathbf{k}\mu}\big)$$
$$\times \Big[ \big(f_{\mathbf{q}+\mathbf{p}\,\nu'}^{\mathrm{el}} - 1\big)\big(f_{\mathbf{k}\mu}^{\mathrm{el}} - 1\big)f_{\mathbf{k}+\mathbf{p}\,\nu}^{\mathrm{el}}f_{\mathbf{q}\mu'}^{\mathrm{el}} - f_{\mathbf{q}+\mathbf{p}\,\nu'}^{\mathrm{el}}f_{\mathbf{k}\mu}^{\mathrm{el}}\big(f_{\mathbf{k}+\mathbf{p}\,\nu}^{\mathrm{el}} - 1\big)\big(f_{\mathbf{q}\mu'}^{\mathrm{el}} - 1\big) \Big], \tag{69}$$

and

$$S_{\mathbf{k}\mu\mu}^{\mathrm{FM}} = 2\pi \sum_{\mathbf{q}\nu\alpha} \frac{\left| g_{\mathbf{q}-\mathbf{k}\alpha,\mu\nu}^{d}(\mathbf{k}) \right|^2}{2\omega_{\mathbf{k}-\mathbf{q}\alpha}}$$
$$\times \Big\{ \delta\big(\epsilon_{\mathbf{q}\nu} - \epsilon_{\mathbf{k}\mu} + \omega_{\mathbf{k}-\mathbf{q}\alpha}\big)\big[\big(f_{\mathbf{q}\nu}^{\mathrm{el}} - 1\big)f_{\mathbf{k}\mu}^{\mathrm{el}}\mathrm{Re}\big[B_{\mathbf{k}-\mathbf{q}\alpha}^{>}\big] - f_{\mathbf{q}\nu}^{\mathrm{el}}\big(f_{\mathbf{k}\mu}^{\mathrm{el}} - 1\big)\mathrm{Re}\big[B_{\mathbf{k}-\mathbf{q}\alpha}^{<}\big]\big]$$
$$+ \delta\big(\epsilon_{\mathbf{q}\nu} - \epsilon_{\mathbf{k}\mu} - \omega_{\mathbf{k}-\mathbf{q}\alpha}\big)\big[\big(f_{\mathbf{q}\nu}^{\mathrm{el}} - 1\big)f_{\mathbf{k}\mu}^{\mathrm{el}}\mathrm{Re}\big[B_{\mathbf{q}-\mathbf{k}\alpha}^{<}\big] - f_{\mathbf{q}\nu}^{\mathrm{el}}\big(f_{\mathbf{k}\mu}^{\mathrm{el}} - 1\big)\mathrm{Re}\big[B_{\mathbf{q}-\mathbf{k}\alpha}^{>}\big]\big] \Big\}. \tag{70}$$

The Fan-Migdal scattering term depends on both phononic occupations and coherences. It is easy to verify that $\sum_{\mathbf{k}\mu} S_{\mathbf{k}\mu\mu}^{\mathrm{GW}}(t) = \sum_{\mathbf{k}\mu} S_{\mathbf{k}\mu\mu}^{\mathrm{FM}}(t) = 0$, which guarantee the conservation of the total number of electrons. We mention that the GKBA gives identical results for both scattering terms.

The off-diagonal scattering term in Eq. (66) is usually simplified as

$$S_{\mathbf{k}\mu\nu}^{\mathrm{el}}(t) \simeq -\Gamma_{\mathbf{k}\mu\nu}^{\mathrm{pol}}(t)p_{\mathbf{k}\mu\nu}(t), \quad \forall\,\mu \neq \nu. \tag{71}$$

In fact, the polarizations carry information on the electronic coherence and are expected to vanish after the photo-excitation. The polarization rates $\Gamma_{\mathbf{k}\mu\nu}^{\mathrm{pol}}$ can be calculated by different means [71–73], although they are often treated as fitting parameters. A semi-empirical way to estimate them is based on the observation that the electronic scattering term in Eq. (68) has the following mathematical structure

$$S_{\mathbf{k}\mu\mu}^{\mathrm{el}}(t) = -2\Gamma_{\mathbf{k}\mu\mu}^{\mathrm{el},>}(t)f_{\mathbf{k}\mu}^{\mathrm{el}}(t) - 2\Gamma_{\mathbf{k}\mu\mu}^{\mathrm{el},<}(t)\big(f_{\mathbf{k}\mu}^{\mathrm{el}}(t) - 1\big). \tag{72}$$

It is easy to verify that $\Gamma_{\mathbf{k}\mu\mu}^{\mathrm{el},\lessgtr}(t) \geq 0$ for vanishing phononic coherences, i.e., $\Theta_{\mathbf{q}\alpha} = 0$. Comparing Eq. (72) with Eq. (37) we infer that our simplifications have led to diagonal electronic scattering rates. Taking the $(\mu, \nu)$ element of Eq. (37) we then obtain the following expression for the polarization rates

$$\Gamma_{\mathbf{k}\mu\nu}^{\mathrm{pol}}(t) = \Gamma_{\mathbf{k}\mu\mu}^{\mathrm{el},>}(t) + \Gamma_{\mathbf{k}\mu\mu}^{\mathrm{el},<}(t) + \Gamma_{\mathbf{k}\nu\nu}^{\mathrm{el},>}(t) + \Gamma_{\mathbf{k}\nu\nu}^{\mathrm{el},<}(t). \tag{73}$$

It is worth remarking that the Markovian approximation of the GKBA equations of motion leads to unphysical polarization rates, see Appendix B.

For a full time-dependent framework of electrons and phonons Eqs. (65) and (66) must be coupled to the equations of motion for the nuclear displacements and phononic density matrix. The treatment of phonons necessitates a preliminary discussion on the equilibrium response function at clamped nuclei. We here consider the RPA $\chi = \chi^0 + \chi^0 \nu \chi = \chi^0 + \chi^0 W \chi^0$. Omitting time integrals and momentum labels, and using for $\chi$ the same index structure as in Fig. 1 we have

$$\chi_{\substack{\mu\mu' \\ \nu\nu'}} = \delta_{\mu\mu'}\delta_{\nu\nu'}\chi^0_{\substack{\mu \\ \nu}} + \chi^0_{\substack{\mu \\ \nu}} W_{\mu\nu'\mu'\nu}\chi^0_{\substack{\mu' \\ \nu'}}, \tag{74}$$

where we take into account that $\chi^0_{\substack{\mu\mu' \\ \nu\nu'}} = \delta_{\mu\mu'}\delta_{\nu\nu'}\chi^0_{\substack{\mu \\ \nu}}$ due to simplification (S1). For a semiconductor at low temperature $\chi^{0,\lessgtr}_{\substack{\mu \\ \nu}} \simeq 0$ if $\mu$ and $\nu$ are both conduction or valence bands. Therefore, the only sizable elements of the response function are those for which the indices of the pairs $(\mu, \mu')$ and $(\nu, \nu')$ are either conduction-valence or valence-conduction.

## 7.2 Nuclear displacements

The equation of motion Eq. (16) for the nuclear displacements contains the elastic tensor $K$. It would be desirable to formulate a simplified equation where the elastic tensor is renormalized by the phononic self-energy, giving rise to the BO frequencies, see Eq. (5). From Eqs. (65) and (66) we infer that only the polarizations $p_{\mathbf{k}\mu\nu}$ with indices $(\mu,\nu)$ either conduction-valence or valence-conduction change linearly with the driving field, since it is only in this case that $f^{\text{el}}_{\mathbf{k}\nu}(0) - f^{\text{el}}_{\mathbf{k}\mu}(0)$ is sizable. All other elements of the electronic density matrix change at least quadratically. We then write the polarizations as

$$p_{\mathbf{k}\mu\nu} = p^{\text{inter}}_{\mathbf{k}\mu\nu} + p^{\text{intra}}_{\mathbf{k}\mu\nu}, \tag{75}$$

where $p^{\text{inter}}_{\mathbf{k}\mu\nu}$ is non vanishing only if $\mu$ is a valence (conduction) band and $\nu$ is a conduction (valence) band whereas $p^{\text{intra}}_{\mathbf{k}\mu\nu}$ is nonvanishing only if $\mu$ and $\nu$ are both valence or conduction bands. Taking into account that $p_{\mathbf{k}\mu\nu}(0) = 0$ we can rewrite the first term in the r.h.s. of Eq. (16) as

$$\sum_{\mathbf{k}\mu\nu} g_{0\alpha,\nu\mu}(\mathbf{k})\Delta\rho^<_{\mathbf{k}\mu\nu} = \sum_{\mathbf{k}\mu\nu} g_{0\alpha,\nu\mu}(\mathbf{k})p^{\text{inter}}_{\mathbf{k}\mu\nu} + \sum_{\mathbf{k}\mu\nu} g_{0\alpha,\nu\mu}(\mathbf{k})\big[\delta_{\mu\nu}\Delta f^{\text{el}}_{\mathbf{k}\mu} + p^{\text{intra}}_{\mathbf{k}\mu\nu}\big]. \tag{76}$$

The third simplification (S3) consists in expressing $p^{\text{inter}}_{\mathbf{k}\mu\nu}$ using the Kubo formula, and in treating the nuclei in the Ehrenfest approximation. Omitting the dependence on momenta we have, see Appendix A,

$$p^{\text{inter}}_{\mu\nu}(t) = \int dt' \sum_{\substack{\mu'\nu' \\ \nu\nu'}} \chi^{\text{R}}_{\substack{\mu\mu' \\ \nu\nu'}}(t,t')\Big[\Omega_{\mu'\nu'}(t') + \sum_\alpha g_{\alpha,\mu'\nu'}U_\alpha(t')\Big], \tag{77}$$

with $\chi$ the response function in the clamped-nuclei approximation. For a response function that decays fast as $|t - t'| \to \infty$ we can evaluate the slowly varying function $U_\alpha$ at time $t$ instead of $t'$, and hence perform the time integral $\int dt' \chi^{\text{R}}(t,t') = \chi^{\text{R}}(\omega = 0)$. Taking into account Eq. (74) and the discussion below it, we then find the following important result

$$\sum_{\mu\nu} g_{\alpha,\nu\mu} p^{\text{inter}}_{\mu\nu} + \sum_{\alpha'} K_{\alpha\alpha'}U_{\alpha'} = \omega^2_\alpha U_\alpha + \int dt' \sum_{\substack{\mu\mu' \\ \nu\nu'}} g_{\alpha,\nu\mu}\chi^{\text{R}}_{\substack{\mu\mu' \\ \nu\nu'}}(t,t')\Omega_{\mu'\nu'}(t'), \tag{78}$$

where we have recognized the phononic self-energy in the clamped-nuclei plus static approximation, see Eq. (50), and used Eq. (5). In conclusion, the equation of motion Eq. (16) becomes (reintroducing the dependence on momenta)

$$\frac{d^2}{dt^2}U_{\mathbf{0}\alpha}(t) + \omega_{\mathbf{0}\alpha}^2 U_{\mathbf{0}\alpha}(t) = -\sum_{\mathbf{k}\mu\nu} g_{\mathbf{0}\alpha,\nu\mu}(\mathbf{k})\big[\delta_{\mu\nu}\Delta f_{\mathbf{k}\mu}^{\text{el}}(t) + p_{\mathbf{k}\mu\nu}^{\text{intra}}(t)\big]$$

$$-\int dt' \sum_{\substack{\mathbf{k}\mathbf{k}' \\ \mu\mu' \\ \nu\nu'}} g_{\mathbf{0}\alpha,\nu\mu}(\mathbf{k})\chi^{\text{R}}_{\mathbf{k}\mathbf{k}',\,\substack{\mu\mu' \\ \nu\nu'}}(\mathbf{0};t,t')\Omega_{\mathbf{k}'\mu'\nu'}(t'). \qquad (79)$$

We emphasize that in Eq. (79) the *e-ph* coupling is the undressed one. We also observe that Eq. (79) differs from the equation of motion $\frac{d^2}{dt^2}U_{\mathbf{0}\alpha} + \omega_{\mathbf{0}\alpha}^2 U_{\mathbf{0}\alpha}(t) = -\sum_{\mathbf{k}\mu\nu} g_{\mathbf{0}\alpha,\nu\mu}(\mathbf{k})\Delta\rho_{\mathbf{k}\mu\nu}(t)$, typical of *e-ph* model Hamiltonians. This equation involves the full density matrix, not just the intra-only elements; moreover the last term in Eq. (79) is missing. The root cause of the discrepancy lies in the unjustified replacement of Eq. (4b) with a set of harmonic oscillators, i.e., $\hat{H}_{0,ph} = \sum_{\mathbf{q}\alpha} \omega_{\mathbf{q}\alpha}\hat{b}_{\mathbf{q}\alpha}^\dagger \hat{b}_{\mathbf{q}\alpha}$. As pointed out in Refs. [1,74], model Hamiltonians suffer from a double renormalization of the phononic frequencies.

## 7.3 Phononic occupations and coherences

The starting point is here the equation of motion Eq. (28) in the basis of the BO normal modes. Using the quasi-particle $2 \times 2$ matrix $Q_{\text{qp}}$ in Eq. (51) we can write

$$\frac{d}{dt}\gamma_{\mathbf{q}\alpha\alpha}^<(t) + \begin{pmatrix} 0 & -1 \\ \omega_{\mathbf{q}\alpha}^2 & 0 \end{pmatrix}\gamma_{\mathbf{q}\alpha\alpha}^<(t) - \gamma_{\mathbf{q}\alpha\alpha}^<(t)\begin{pmatrix} 0 & -\omega_{\mathbf{q}\alpha}^2 \\ 1 & 0 \end{pmatrix} = S_{\mathbf{q}\alpha\alpha}^{\text{ph}}(t), \qquad (80)$$

where

$$S_{\mathbf{q}\alpha\alpha}^{\text{ph}}(t) \equiv \mathcal{J}\int^t dt'\big[\Pi_{\mathbf{q}\alpha\alpha}^>(t,t')D_{\mathbf{q}\alpha\alpha}^<(t',t) - \Pi_{\mathbf{q}\alpha\alpha}^<(t,t')D_{\mathbf{q}\alpha\alpha}^>(t',t)\big] + \text{h.c.}, \qquad (81)$$

is the *phononic scattering term*. Let us inspect the elements of the $2 \times 2$ matrix $S^{\text{ph}}$. Naming the integral in Eq. (81) with the letter $I$ – hence $I$ is a $2 \times 2$ matrix – and taking into account that the self-energy has only one nonvanishing element, which is the $(1,1)$, we find

$$S^{\text{ph}} = \mathcal{J}I + \text{h.c.} = \begin{pmatrix} 0 & i \\ -i & 0 \end{pmatrix}\begin{pmatrix} I^{11} & I^{12} \\ 0 & 0 \end{pmatrix} + \text{h.c.} = \begin{pmatrix} 0 & iI^{11*} \\ -iI^{11} & -iI^{12} + iI^{12*} \end{pmatrix}.$$

Thus, to calculate $S_{\mathbf{q}\alpha\alpha}^{\text{ph}}$ we only need the $(1,1)$ and $(1,2)$ elements of the phononic GF, whose MGKBA form has been derived in Eqs. (58) and (59). Evaluating the phononic self-energy, see Eq. (56), at the MGKBA GFs, and implementing the Markov approximation (S3) we obtain

$$S_{\mathbf{q}\alpha\alpha}^{21,\text{ph}} = -i\pi\sum_{\mathbf{k}\mu\nu}\frac{|g_{\mathbf{q}\alpha,\mu\nu}^d(\mathbf{k})|^2}{2\omega_{\mathbf{q}\alpha}}\Big\{\delta\big(\epsilon_{\mathbf{q}+\mathbf{k}\nu} - \epsilon_{\mathbf{k}\mu} - \omega_{\mathbf{q}\alpha}\big)\big[(f_{\mathbf{q}+\mathbf{k}\nu}^{\text{el}} - 1)f_{\mathbf{k}\mu}^{\text{el}}B_{\mathbf{q}\alpha}^< - f_{\mathbf{q}+\mathbf{k}\nu}^{\text{el}}(f_{\mathbf{k}\mu}^{\text{el}} - 1)B_{\mathbf{q}\alpha}^>\big]$$

$$+ \delta\big(\epsilon_{\mathbf{q}+\mathbf{k}\nu} - \epsilon_{\mathbf{k}\mu} + \omega_{\mathbf{q}\alpha}\big)\big[(f_{\mathbf{q}+\mathbf{k}\nu}^{\text{el}} - 1)f_{\mathbf{k}\mu}^{\text{el}}B_{-\mathbf{q}\alpha}^{>*} - f_{\mathbf{q}+\mathbf{k}\nu}^{\text{el}}(f_{\mathbf{k}\mu}^{\text{el}} - 1)B_{-\mathbf{q}\alpha}^{<*}\big]\Big\}, \qquad (82)$$

and

$$S_{\mathbf{q}\alpha\alpha}^{22,\text{ph}} = \pi\sum_{\mathbf{k}\mu\nu}\frac{|g_{\mathbf{q}\alpha,\mu\nu}^d(\mathbf{k})|^2}{2}\Big\{\delta\big(\epsilon_{\mathbf{q}+\mathbf{k}\nu} - \epsilon_{\mathbf{k}\mu} - \omega_{\mathbf{q}\alpha}\big)\big[(f_{\mathbf{q}+\mathbf{k}\nu}^{\text{el}} - 1)f_{\mathbf{k}\mu}^{\text{el}}C_{\mathbf{q}\alpha}^< - f_{\mathbf{q}+\mathbf{k}\nu}^{\text{el}}(f_{\mathbf{k}\mu}^{\text{el}} - 1)C_{\mathbf{q}\alpha}^>\big]$$

$$- \delta\big(\epsilon_{\mathbf{q}+\mathbf{k}\nu} - \epsilon_{\mathbf{k}\mu} + \omega_{\mathbf{q}\alpha}\big)\big[(f_{\mathbf{q}+\mathbf{k}\nu}^{\text{el}} - 1)f_{\mathbf{k}\mu}^{\text{el}}C_{-\mathbf{q}\alpha}^{>*} - f_{\mathbf{q}+\mathbf{k}\nu}^{\text{el}}(f_{\mathbf{k}\mu}^{\text{el}} - 1)C_{-\mathbf{q}\alpha}^{<*}\big]\Big\} + \text{h.c.}, \qquad (83)$$

and the more obvious ones

$$S_{\mathbf{q}\alpha\alpha}^{11,\text{ph}} = 0,\tag{84}$$

$$S_{\mathbf{q}\alpha\alpha}^{12,\text{ph}} = [S_{\mathbf{q}\alpha\alpha}^{21,\text{ph}}]^*.\tag{85}$$

With these preliminary results we can construct the equations of motion for the phononic occupations and coherences. From Eqs. (22) we have

$$f_{\mathbf{q}\alpha}^{\text{ph}} = \frac{1}{2}\Big[\omega_{\mathbf{q}\alpha}\gamma_{\mathbf{q}\alpha\alpha}^{11,<} + \frac{1}{\omega_{\mathbf{q}\alpha}}\gamma_{\mathbf{q}\alpha\alpha}^{22,<} - i\big(\gamma_{\mathbf{q}\alpha\alpha}^{12,<} - \gamma_{\mathbf{q}\alpha\alpha}^{21,<}\big)\Big],\tag{86a}$$

$$\Theta_{\mathbf{q}\alpha} = \frac{1}{2}\Big[\omega_{\mathbf{q}\alpha}\gamma_{\mathbf{q}\alpha\alpha}^{11,<} - \frac{1}{\omega_{\mathbf{q}\alpha}}\gamma_{\mathbf{q}\alpha\alpha}^{22,<} + i\big(\gamma_{\mathbf{q}\alpha\alpha}^{12,<} + \gamma_{\mathbf{q}\alpha\alpha}^{21,<}\big)\Big].\tag{86b}$$

Using Eq. (80) we then find

$$\frac{d}{dt}f_{\mathbf{q}\alpha}^{\text{ph}} = S_{\mathbf{q}\alpha\alpha}^{\text{ph}-\text{occ}},\tag{87a}$$

$$\frac{d}{dt}\Theta_{\mathbf{q}\alpha} + 2i\omega_{\mathbf{q}\alpha}\Theta_{\mathbf{q}\alpha} = S_{\mathbf{q}\alpha\alpha}^{\text{ph}-\text{coh}},\tag{87b}$$

where $S_{\mathbf{q}\alpha\alpha}^{\text{ph}-\text{occ}} \equiv \frac{1}{2}\Big[\omega_{\mathbf{q}\alpha}S_{\mathbf{q}\alpha\alpha}^{11,\text{ph}} + \frac{1}{\omega_{\mathbf{q}\alpha\alpha}}S_{\mathbf{q}\alpha\alpha}^{22,\text{ph}} - i\big(S_{\mathbf{q}\alpha\alpha}^{12,\text{ph}} - S_{\mathbf{q}\alpha\alpha}^{21,\text{ph}}\big)\Big]$ reads

$$
\begin{aligned}
S_{\mathbf{q}\alpha\alpha}^{\text{ph}-\text{occ}} = 2\pi\sum_{\mathbf{k}\mu\nu}\frac{|g_{\mathbf{q}\alpha,\mu\nu}^{d}(\mathbf{k})|^2}{2\omega_{\mathbf{q}\alpha}}\\
= \Big\{\delta\big(\epsilon_{\mathbf{q}+\mathbf{k}\nu} - \epsilon_{\mathbf{k}\mu} - \omega_{\mathbf{q}\alpha}\big)\Big[\big(f_{\mathbf{q}+\mathbf{k}\nu}^{\text{el}} - 1\big)f_{\mathbf{k}\mu}^{\text{el}}f_{\mathbf{q}\alpha}^{\text{ph}} - f_{\mathbf{q}+\mathbf{k}\nu}^{\text{el}}\big(f_{\mathbf{k}\mu}^{\text{el}} - 1\big)\big(f_{\mathbf{q}\alpha}^{\text{ph}} + 1\big)\Big]\\
+ \delta\big(\epsilon_{\mathbf{q}+\mathbf{k}\nu} - \epsilon_{\mathbf{k}\mu} + \omega_{\mathbf{q}\alpha}\big)\Big[\big(f_{\mathbf{q}+\mathbf{k}\nu}^{\text{el}} - 1\big)f_{\mathbf{k}\mu}^{\text{el}} - f_{\mathbf{q}+\mathbf{k}\nu}^{\text{el}}\big(f_{\mathbf{k}\mu}^{\text{el}} - 1\big)\Big]\text{Re}\big[\Theta_{\mathbf{q}\alpha}\big]\Big\},
\end{aligned}\tag{88}
$$

and $S_{\mathbf{q}\alpha\alpha}^{\text{ph}-\text{coh}} \equiv \frac{1}{2}\Big[\omega_{\mathbf{q}\alpha\alpha}S_{\mathbf{q}\alpha\alpha}^{11,\text{ph}} - \frac{1}{\omega_{\mathbf{q}\alpha\alpha}}S_{\mathbf{q}\alpha\alpha}^{22,\text{ph}} + i\big(S_{\mathbf{q}\alpha\alpha}^{12,\text{ph}} + S_{\mathbf{q}\alpha\alpha}^{21,\text{ph}}\big)\Big]$ reads

$$
\begin{aligned}
S_{\mathbf{q}\alpha\alpha}^{\text{ph}-\text{coh}} = \pi\sum_{\mathbf{k}\mu\nu}\frac{|g_{\mathbf{q}\alpha,\mu\nu}^{d}(\mathbf{k})|^2}{2\omega_{\mathbf{q}\alpha}}\\
\times\Big\{\delta\big(\epsilon_{\mathbf{q}+\mathbf{k}\nu} - \epsilon_{\mathbf{k}\mu} - \omega_{\mathbf{q}\alpha}\big)\\
\times\Big[\big(f_{\mathbf{q}+\mathbf{k}\nu}^{\text{el}} - 1\big)f_{\mathbf{k}\mu}^{\text{el}}\big(\Theta_{\mathbf{q}\alpha} - f_{\mathbf{q}\alpha}^{\text{ph}}\big) - f_{\mathbf{q}+\mathbf{k}\nu}^{\text{el}}\big(f_{\mathbf{k}\mu}^{\text{el}} - 1\big)\big(\Theta_{\mathbf{q}\alpha} - f_{\mathbf{q}\alpha}^{\text{ph}} - 1\big)\Big]\\
+ \delta\big(\epsilon_{\mathbf{q}+\mathbf{k}\nu} - \epsilon_{\mathbf{k}\mu} + \omega_{\mathbf{q}\alpha}\big)\\
\times\Big[\big(f_{\mathbf{q}+\mathbf{k}\nu}^{\text{el}} - 1\big)f_{\mathbf{k}\mu}^{\text{el}}\big(f_{-\mathbf{q}\alpha}^{\text{ph}} + 1 - \Theta_{\mathbf{q}\alpha}\big) - f_{\mathbf{q}+\mathbf{k}\nu}^{\text{el}}\big(f_{\mathbf{k}\mu}^{\text{el}} - 1\big)\big(f_{-\mathbf{q}\alpha}^{\text{ph}} - \Theta_{\mathbf{q}\alpha}\big)\Big]\Big\}.
\end{aligned}\tag{89}
$$

The equation of motion for the phononic coherences deserves further investigation. Let us write $\Theta_{\mathbf{q}\alpha} = \Theta_{\mathbf{q}\alpha}^{(r)} + i\Theta_{\mathbf{q}\alpha}^{(i)}$ as the sum of its real and imaginary part. Then Eqs. (87b) and (89) imply

$$\frac{d}{dt}\Theta_{\mathbf{q}\alpha}^{(i)} + 2\omega_{\mathbf{q}\alpha}\Theta_{\mathbf{q}\alpha}^{(r)} = -\Gamma_{\mathbf{q}\alpha}^{\text{coh}}\Theta_{\mathbf{q}\alpha}^{(i)},\tag{90a}$$

$$\frac{d}{dt}\Theta_{\mathbf{q}\alpha}^{(r)} - 2\omega_{\mathbf{q}\alpha}\Theta_{\mathbf{q}\alpha}^{(i)} = -\Gamma_{\mathbf{q}\alpha}^{\text{coh}}\Theta_{\mathbf{q}\alpha}^{(r)} + S_{\mathbf{q}\alpha\alpha}^{\text{ph}-\text{coh}}\Big|_{\Theta_{\mathbf{q}\alpha}=0},\tag{90b}$$

where

$$\Gamma_{\mathbf{q}\alpha}^{\text{coh}} = \pi\sum_{\mathbf{k}\mu\nu}\frac{|g_{\mathbf{q}\alpha,\mu\nu}^{d}(\mathbf{k})|^2}{2\omega_{\mathbf{q}\alpha}}\Big[\delta\big(\epsilon_{\mathbf{q}+\mathbf{k}\nu} - \epsilon_{\mathbf{k}\mu} - \omega_{\mathbf{q}\alpha}\big) - \delta\big(\epsilon_{\mathbf{q}+\mathbf{k}\nu} - \epsilon_{\mathbf{k}\mu} + \omega_{\mathbf{q}\alpha}\big)\Big]\Big[f_{\mathbf{k}\mu}^{\text{el}} - f_{\mathbf{q}+\mathbf{k}\nu}^{\text{el}}\Big].\tag{91}$$

Table 1: Table summarizing the SEPE. The electronic scattering term is given by the sum of Eq. (69) and Eq. (70) whereas the phononic scattering terms are given by Eqs. (88) and (89).

| Definition | Equations |
|---|---|
| Electronic occupation | $\frac{d}{dt}f^{\text{el}}_{\mathbf{k}\mu} + i\sum_{\nu\neq\mu}\left(\Omega^{\text{ren}}_{\mathbf{k}\mu\nu}p_{\mathbf{k}\nu\mu} - p_{\mathbf{k}\mu\nu}\Omega^{\text{ren}}_{\mathbf{k}\nu\mu}\right) = S^{\text{el}}_{\mathbf{k}\mu\mu}$ |
| Electronic polarization | $\frac{d}{dt}p_{\mathbf{k}\mu\nu} + i(\tilde{\epsilon}_{\mathbf{k}\mu} - \tilde{\epsilon}_{\mathbf{k}\nu})p_{\mathbf{k}\mu\nu} + i\Omega^{\text{ren}}_{\mathbf{k}\mu\nu}\left(f^{\text{el}}_{\mathbf{k}\nu} - f^{\text{el}}_{\mathbf{k}\mu}\right)$ $+ i\sum_{\nu'\neq\nu}\Omega^{\text{ren}}_{\mathbf{k}\mu\nu'}p_{\mathbf{k}\nu'\nu} - i\sum_{\nu'\neq\mu}p_{\mathbf{k}\mu\nu'}\Omega^{\text{ren}}_{\mathbf{k}\nu'\nu} = -\Gamma^{\text{pol}}_{\mathbf{k}\mu\nu}p_{\mathbf{k}\mu\nu}$ |
| Nuclear displacement | $\frac{d^2}{dt^2}U_{\mathbf{0}\alpha} + \omega^2_{\mathbf{0}\alpha}U_{\mathbf{0}\alpha} = -\sum_{\mathbf{k}\mu\nu}g_{\mathbf{0}\alpha,\nu\mu}(\mathbf{k})\left[\delta_{\mu\nu}\Delta f^{\text{el}}_{\mathbf{k}\mu} + p^{\text{intra}}_{\mathbf{k}\mu\nu}\right]$ $- \int dt' \sum_{\substack{\mathbf{k}\mathbf{k}' \\ \mu\mu' \\ \nu\nu'}} g_{\mathbf{0}\alpha,\nu\mu}(\mathbf{k})\chi^{\text{R}}_{\mathbf{k}\mathbf{k}',\,\substack{\mu\mu' \\ \nu\nu'}}(\mathbf{0};t,t')\Omega_{\mathbf{k}'\mu'\nu'}(t')$ |
| Phonon occupation | $\frac{d}{dt}f^{\text{ph}}_{\mathbf{q}\alpha} = S^{\text{ph}-\text{occ}}_{\mathbf{q}\alpha\alpha}$ |
| Phonon coherence | $\frac{d}{dt}\Theta_{\mathbf{q}\alpha} + 2i\omega_{\mathbf{q}\alpha}\Theta_{\mathbf{q}\alpha} = S^{\text{ph}-\text{coh}}_{\mathbf{q}\alpha\alpha}$ |
| Quasiparticle energy | $\tilde{\epsilon}_{\mathbf{k}\mu}(t) = \epsilon_{\mathbf{k}\mu} + \sum_{\mathbf{k}'\mu'\nu'}\left(v_{\mathbf{k}\mu\mathbf{k}'\mu'\mathbf{k}'\nu'\mathbf{k}\mu} - W_{\mathbf{k}\mu\mathbf{k}'\mu'\mathbf{k}\mu\mathbf{k}'\nu'}\right)\left[\rho^<_{\mathbf{k}'\nu'\mu'}(t) - \rho^<_{\mathbf{k}'\nu'\mu'}(0)\right]$ |
| Renorm. Rabi frequency | $\Omega^{\text{ren}}_{\mathbf{k}\mu\nu} = \Omega_{\mathbf{k}\mu\nu} + \Delta\Sigma^\delta_{\mathbf{k}\mu\nu} + \sum_\alpha g_{\mathbf{0}\alpha,\mu\nu}(\mathbf{k})U_{\mathbf{0}\alpha}$ |

The *coherence rate* $\Gamma^{\text{coh}}_{\mathbf{q}\alpha}$ is positive when the electronic occupations satisfy the inequality $f_{\mathbf{k}\mu} > f_{\mathbf{q}+\mathbf{k}\nu}$ for $\epsilon_{\mathbf{q}+\mathbf{k}\nu} > \epsilon_{\mathbf{k}\mu}$ and indices $(\mu, \nu)$ that are both conduction or both valence bands (contributions with $\mu$ a valence index and $\nu$ a conduction index or viceversa vanish due to the Dirac delta); this condition holds true for a quasi-thermalized distribution of carriers. We prove in Section 7.5 that $S^{\text{ph}-\text{coh}}_{\mathbf{q}\alpha\alpha}\Big|_{\Theta_{\mathbf{q}\alpha}=0}$ vanishes for thermal electronic and phononic occupations, and it is therefore small for occupations close to thermal ones. Ignoring this term in Eq. (90b) and assuming $\Gamma^{\text{coh}}_{\mathbf{q}\alpha}(t)$ weakly dependent on time, the most general solution for the phononic coherences is

$$\Theta^{(i)}_{\mathbf{q}\alpha}(t) = \Theta_{0,\mathbf{q}\alpha}\cos\left(2\omega_{\mathbf{q}\alpha}t + \phi_{0,\mathbf{q}\alpha}\right)e^{-\Gamma^{\text{coh}}_{\mathbf{q}\alpha}t}, \tag{92a}$$

$$\Theta^{(r)}_{\mathbf{q}\alpha}(t) = \Theta_{0,\mathbf{q}\alpha}\sin\left(2\omega_{\mathbf{q}\alpha}t + \phi_{0,\mathbf{q}\alpha}\right)e^{-\Gamma^{\text{coh}}_{\mathbf{q}\alpha}t}. \tag{92b}$$

It is worth remarking that the Markovian approximation of the GKBA equations of motion for the phononic coherences differs from the MGKBA equation Eqs. (90) in that the sign of $\Gamma^{\text{coh}}_{\mathbf{q}\alpha}$ is reversed, see Appendix B. This implies that the equilibrium solution is unstable in the GKBA version of the SEPE.

## 7.4 Short and long driving

In Table 1 we summarize the SEPE for the electronic and phononic degrees of freedom. The electronic scattering term is given by the sum of Eq. (69) – *GW* – and Eq. (70) – Fan-Migdal

– whereas the phononic scattering terms are given by Eqs. (88) and (89). All scattering terms except the *GW* one depend on both phononic occupations and coherences.

If the duration $T_{\text{drive}}$ of the driving field is much shorter than a typical phonon period (for a phonon frequency $\lesssim 100$ meV the phonon period is $\gtrsim 40$ fs) then the nuclei remain essentially still while the field is on. For such short drivings we can neglect the last term in the third equation of Table 1, as it grows linearly with $T_{\text{drive}}$. The energy shift $\sum_\alpha g_{0\alpha,\mu\nu}(\mathbf{k})U_{0\alpha}$ in $\Omega^{\text{ren}}_{\mathbf{k}\mu\nu}$, see Eq. (67), is typically of the order of a few meV and it is responsible for time-dependent modulations in the optical spectra [33–37]. To capture this effect it is crucial to use $\Omega^{\text{ren}}_{\mathbf{k}\mu\nu}$ in the two terms that multiply the polarizations in the second equation of Table 1. All other $\Omega^{\text{ren}}_{\mathbf{k}\mu\nu}$ can approximated with $\Omega_{\mathbf{k}\mu\nu}$.

For long driving fields, i.e., $T_{\text{drive}}$ a few hundreds of fs or longer, the last term in the third equation of Table 1 cannot be discarded. In this case the nuclei are expected to slowly attain new positions and no time-dependent modulations of the optical spectra are to be expected. The nuclear shifts are mainly responsible for a few meV renormalization of the quasi-particle energies. Thus, if our focus is solely on occupations and coherences, or if we do not require a meV resolution of the optical spectra then we can solve the SEPE with $U_{0\alpha} = 0$.

## 7.5 Equilibrium and steady-state solutions

Let us discuss the stationary solutions of the SEPE. We assign a chemical potential $\mu_c$ to all conduction bands and $\mu_v$ to all valence bands, respectively, and show that all scattering terms vanish if the electronic occupations $f^{\text{el}}_{\mathbf{k}\nu} = 1/[e^{\beta(\epsilon_{\mathbf{k}\nu}-\mu_\nu)}+1]$ (noninteracting finite-temperature electrons), the phononic occupations $f^{\text{ph}}_{\mathbf{q}\alpha} = 1/[e^{\beta\omega_{\mathbf{q}\alpha}}-1]$ (noninteracting finite-temperature phonons), and the phononic coherences $\Theta_{\mathbf{q}\alpha} = 0$. Let us consider the *GW* scattering term in Eq. (69). Taking into account that $f^{\text{el}}_{\mathbf{k}\nu}/[1-f^{\text{el}}_{\mathbf{k}\nu}] = e^{-\beta(\epsilon_{\mathbf{k}\nu}-\mu_\nu)}$, the energy conservation enforced by the Dirac delta implies

$$\frac{f^{\text{el}}_{\mathbf{k}+\mathbf{p}\nu} f^{\text{el}}_{\mathbf{q}\mu'} e^{\beta(\mu_{\mu'}+\mu_\nu)}}{(f^{\text{el}}_{\mathbf{k}+\mathbf{p}\nu}-1)(f^{\text{el}}_{\mathbf{q}\mu'}-1)} = \frac{f^{\text{el}}_{\mathbf{q}+\mathbf{p}\nu'} f^{\text{el}}_{\mathbf{k}\mu} e^{\beta(\mu_{\nu'}+\mu_\mu)}}{(f^{\text{el}}_{\mathbf{q}+\mathbf{p}\nu'}-1)(f^{\text{el}}_{\mathbf{k}\mu}(t)-1)},$$

and therefore all terms in $S^{\text{GW}}_{\mathbf{k}\mu\mu}$ with $\mu_{\mu'}+\mu_\nu = \mu_{\nu'}+\mu_\mu$ vanish. The only case for which $\mu_{\mu'}+\mu_\nu \neq \mu_{\nu'}+\mu_\mu$ is when $(\mu',\nu)$ are both conduction (valence) bands and $(\nu',\mu)$ are both valence (conduction) bands. However, for this choice of indices the argument of the Dirac delta is at least twice the quasi-particle gap. We conclude that $S^{\text{GW}}_{\mathbf{k}\mu\mu} = 0$.

Similarly, for the Fan-Migdal scattering term in Eq. (70) we have

$$\frac{f^{\text{el}}_{\mathbf{k}\mu}}{f^{\text{el}}_{\mathbf{k}\mu}-1}\left[f^{\text{ph}}_{\mathbf{k}-\mathbf{q}\alpha}+1\right] = e^{-\beta(\mu_\nu-\mu_\mu)}\frac{f^{\text{el}}_{\mathbf{q}\nu}}{f^{\text{el}}_{\mathbf{q}\nu}-1}f^{\text{ph}}_{\mathbf{k}-\mathbf{q}\alpha},$$

where we enforce the energy conservation $\epsilon_{\mathbf{k}\mu} = \epsilon_{\mathbf{q}\nu}+\omega_{\mathbf{k}-\mathbf{q}\alpha}$. A similar relation can be derived for the term with $\epsilon_{\mathbf{k}\mu} = \epsilon_{\mathbf{q}\nu}-\omega_{\mathbf{k}-\mathbf{q}\alpha}$. Therefore all terms in $S^{\text{FM}}_{\mathbf{k}\mu\mu}$ with $\mu_\nu = \mu_\mu$ vanish. The only case for which $\mu_\nu \neq \mu_\mu$ is when $\nu$ is a conduction (valence) band and $\mu$ is a valence (conduction) band. However, for this choice of indices the argument of the Dirac delta is about the quasi-particle gap. We conclude that $S^{\text{FM}}_{\mathbf{k}\mu\mu} = 0$. The same arguments can be used to show that $S^{\text{ph-occ}}_{\mathbf{q}\alpha\alpha} = S^{\text{ph-coh}}_{\mathbf{q}\alpha\alpha} = 0$.

The most general steady-state solution of the SEPE with $\Omega^{\text{ren}}_{\mathbf{k}\mu\nu} = 0$ is given by noninteracting electronic and phononic occupations *at the same temperature*, and vanishing electronic polarizations and phononic coherences. Among all these solutions there exists the equilibrium one, where $\beta$ is the equilibrium inverse temperature and $\mu_c = \mu_v = \mu$ is the equilibrium chemical potential. The steady-state solution, if attained, is expected to have a temperature higher

than the equilibrium temperature since the external field injects energy in the system. This means that $\Delta f_{\mathbf{k}\mu}^{\text{el}} = f_{\mathbf{k}\mu}^{\text{el}}(t \to \infty) - f_{\mathbf{k}\mu}^{\text{el}}(t=0)$ is, in general, different from zero, and therefore the nuclei attain new positions $U_{\mathbf{0}\alpha}(t \to \infty) = -\frac{1}{\omega_{\mathbf{0}\alpha}^2}\sum_{\mathbf{k}\mu} g_{\mathbf{0}\alpha,\mu\mu}(\mathbf{k})\Delta f_{\mathbf{k}\mu}^{\text{el}}$, see Eq. (79). These displacements can be either negative or positive, depending on the sign and magnitude of the *e-ph* couplings. Because of the non-zero nuclear displacements, $\Omega_{\mathbf{k}\mu\nu}^{\text{ren}}(t \to \infty)$ is small but not exactly zero. Consequently, the steady-state occupations, polarizations, and coherences differ slightly from the thermal values.

## 8  Recovering the semiconductor Bloch and Boltzmann equations

Historically, the SBE exclusively addressed the electron dynamics [14, 75], implicitly assuming that the phonons remained in thermal equilibrium [59, 60, 76]. These equations follow from the SEPE by setting $U_{\mathbf{0},\alpha} = \Theta_{\mathbf{q}\alpha} = 0$ and $f_{\mathbf{q}\alpha}^{\text{ph}} = 1/[e^{\beta\omega_{\mathbf{q}\alpha}} - 1]$, thus only the first two equations in Table 1 need to be solved. Improvements of the SBE in which the phononic occupations $f_{\mathbf{q}\alpha}^{\text{ph}}$ satisfy their own equation of motion (fourth equation in Table 1) have also been considered [77, 78].

The SBE simplify further in the so called *incoherent regime*. For times long after the perturbation caused by the external field it is reasonable to assume that the system is well described by a many-body density matrix of the form $\hat{\rho}(t) = \sum_k w_k(t)|\Psi_k\rangle\langle\Psi_k|$, where the many-body states $|\Psi_k\rangle$ have a well defined number of electrons and phonons or, equivalently, are eigenstates of the electronic and phononic number operators

$$\hat{n}_{\mathbf{k}\mu}^{\text{el}}|\Psi_k\rangle = \hat{d}_{\mathbf{k}\mu}^\dagger\hat{d}_{\mathbf{k}\mu}|\Psi_k\rangle = n_{\mathbf{k}\mu}^{\text{el}}|\Psi_k\rangle\,, \qquad n_{\mathbf{k}\mu}^{\text{el}} = 0, 1\,, \tag{93a}$$

$$\hat{n}_{\mathbf{q}\alpha}^{\text{ph}}|\Psi_k\rangle = \hat{b}_{\mathbf{q}\alpha}^\dagger\hat{b}_{\mathbf{q}\alpha}|\Psi_k\rangle = n_{\mathbf{q}\alpha}^{\text{ph}}|\Psi_k\rangle\,, \qquad n_{\mathbf{q}\alpha}^{\text{ph}} = 0, 1, 2, \ldots \tag{93b}$$

In the incoherent regime we have $p_{\mathbf{k}\mu\nu}(t) = U_{\mathbf{0}\alpha} = \Theta_{\mathbf{q}\nu} = 0$, thus only the equations of motion for $f_{\mathbf{k}\mu}^{\text{el}}$ and $f_{\mathbf{q}\alpha}^{\text{ph}}$ (first and fourth equations in Table 1) need to be solved. These are the BE [20, 79–83]. The BE do not account for the interaction between electrons and transverse fields. The nonequilibrium distribution of electrons and phonons enters as the initial value of $f_{\mathbf{k}\mu}^{\text{el}}$ and $f_{\mathbf{q}\alpha}^{\text{ph}}$. In Appendix C we prove that the total entropy of the coupled electron-phonon system is an increasing function of time.

We conclude by observing that the SBE and BE can be derived from the GKBA only if we *impose* that $\Theta_{\mathbf{q}\alpha} = 0$. In fact, $\Theta_{\mathbf{q}\alpha} = 0$ is not a stable solution in GKBA, see Appendix B.

## 9  Conclusions and Outlook

Based on our recent work on the *ab initio* many-body theory of electrons and phonons [1] we derive a simplified set of equations of motion for the electronic occupations and polarizations, nuclear displacements as well as phononic occupations and coherences. We can succinctly summarize the underlying approximations and simplifications as follows: (i) *GW* plus Fan-Migdal self-energy, (ii) MGKBA (iii) diagonal density matrices and retarded GF (S1), (iv) Markov approximation (S2), and (v) Kubo formula for the interband polarization (S3).

By explicitly including the laser field, it is possible to create a nonequilibrium population of electrons and phonons, while simultaneously transferring the laser coherence to both electrons and the nuclear lattice. Currently available electronic structure codes can be used to calculate the electronic and phononic band structures, screened Coulomb interaction $W$ and *e-ph* coupling $g$, which are the only necessary ingredients to run first-principles SEPE simulations.

In particular, the SEPE equations can be readily implemented in SBE and BE codes through a minor change in the Fan-Migdal scattering term, i.e., $f_{\mathbf{q}}^{\mathrm{ph}} \to f_{\mathbf{q}}^{\mathrm{ph}} + \Theta_{\mathbf{q}\alpha}$, and by adding the equations of motion for the nuclear displacements and phononic coherences. These new features pave the way for first-principles studies of the coupling between coherent phonons and excitons as well as squeezed phonon states and time-dependent Debye-Waller factors. Whether the consistent treatment of phononic occupations and coherences has also an impact on the coherent-to-incoherent crossover [64,84] and on the thermalization of electrons and phonons remain to be seen case by case.

On a more fundamental level our work sheds light on the SBE and BE as it clarifies how to derive these methods from the *ab initio* KBE. The mirrored version of the GKBA has been essential in this endeavour. In fact, the GKBA yields unphysical polarization rates and an exponentially diverging solution for the phononic coherences. On the contrary, in MGKBA the equations of motion for the electronic and phononic density matrices have the same structure as a rate equation, and the rates naturally turn out to be positive once the Markovian approximation is made.

We conclude by outlining possible relevant future directions. Following the strategy presented in this work it would be interesting to derive the scattering terms arising from anharmonic effects, e.g., the *ph-ph* interaction [18], which are expected to play a role in the thermalization of the lattice [82]. Another important contribution would be the derivation of an effective equation for the exciton-phonon dynamics [85–88]. We anticipate the appearance of phononic coherences in both cases. Finally, the MGKBA for phonons can equally be used to deal with quantized photons, thus offering an alternative method to study correlation effects and coherences in cavity quantum electrodynamics materials [89,90].

## Acknowledgments

We acknowledge useful discussion with Yaroslav Pavlyukh.

**Funding information** This work has been supported by MIUR PRIN (Grant No. 20173B72NB), MIUR PRIN (Grant No. 2022WZ8LME), INFN through the TIME2QUEST project and Tor Vergata University through Project TESLA.

## A  Response function in the Ehrenfest approximation

Let us denote by $\tilde{\chi}$ the full response function of the electron-phonon system. The diagrammatic expansion of $\tilde{\chi}$ contains diagrams with both *e-e* and *e-ph* interactions. To first order in the external field $\Omega$ the change $\Delta\rho$ in the electronic density matrix is given by (omitting the dependence on momenta)

$$\Delta\rho_{\mu\nu}(t) = \int dt' \sum_{\substack{\mu'\nu' \\ \nu\nu'}} \tilde{\chi}^{\mathrm{R}}_{\substack{\mu\mu' \\ \nu\nu'}}(t,t') \Omega_{\mu'\nu'}(t'). \tag{A.1}$$

The Ehrenfest approximation to the response function is

$$\tilde{\chi}^{\mathrm{R}}_{\substack{\mu\mu' \\ \nu\nu'}}(t,t') = \chi^{\mathrm{R}}_{\substack{\mu\mu' \\ \nu\nu'}}(t,t') + \sum_{\substack{\alpha\beta \\ \mu\mu' \\ \nu\nu'}} \int dt_1 dt_2 \, \chi^{\mathrm{R}}_{\substack{\mu\rho \\ \nu\sigma}}(t,t_1) g_{\alpha,\rho\sigma} D^{11,\mathrm{R}}_{\alpha\beta}(t_1,t_2) g_{\beta,\sigma'\rho'} \tilde{\chi}^{\mathrm{R}}_{\substack{\rho'\mu' \\ \sigma'\nu'}}(t_2,t'), \tag{A.2}$$

where $\chi$ is the response function at clamped nuclei. Substituting Eq. (A.2) into Eq. (A.1) and taking into account that [1]

$$U_\alpha(t_1) = \sum_{\beta\rho'\sigma'} \int dt_2 D_{\alpha\beta}^{11,R}(t_1,t_2) g_{\alpha,\sigma'\rho'} \Delta\rho_{\rho'\sigma'}(t_2), \tag{A.3}$$

we find Eq. (77).

## B  Markovian limit of the GKBA equations

In GKBA the electronic and phononic lesser GFs become [compare with Eqs. (57), (58) and (59)]

$$G_{\mathbf{k}\mu\mu}^<(t,t') = ie^{-i\epsilon_{\mathbf{k}\mu}(t-t')}\left[\theta(t-t')f_{\mathbf{k}\mu}^{el}(t') + \theta(t'-t)f_{\mathbf{k}\mu}^{el}(t)\right], \tag{B.1}$$

$$D_{\mathbf{q}\alpha\alpha}^{11,\lessgtr}(t,t') = \frac{\theta(t-t')}{2i\omega_{\mathbf{q}\alpha}}\left[B_{\mathbf{q}\alpha}^{\lessgtr}(t')e^{-i\omega_{\mathbf{q}\alpha}(t-t')} + B_{-\mathbf{q}\alpha}^{\gtrless*}(t')e^{i\omega_{\mathbf{q}\alpha}(t-t')}\right]$$
$$+ \frac{\theta(t'-t)}{2i\omega_{\mathbf{q}\alpha}}\left[B_{\mathbf{q}\alpha}^{\lessgtr*}(t)e^{-i\omega_{\mathbf{q}\alpha}(t-t')} + B_{-\mathbf{q}\alpha}^{\gtrless}(t)e^{i\omega_{\mathbf{q}\alpha}(t-t')}\right], \tag{B.2}$$

$$D_{\mathbf{q}\alpha\alpha}^{12,\lessgtr}(t,t') = \frac{\theta(t-t')}{2}\left[C_{\mathbf{q}\alpha}^{\lessgtr}(t')e^{-i\omega_{\mathbf{q}\alpha}(t-t')} - C_{-\mathbf{q}\alpha}^{\gtrless*}(t')e^{i\omega_{\mathbf{q}\alpha}(t-t')}\right]$$
$$+ \frac{\theta(t'-t)}{2}\left[B_{\mathbf{q}\alpha}^{\lessgtr*}(t)e^{-i\omega_{\mathbf{q}\alpha}(t-t')} - B_{-\mathbf{q}\alpha}^{\gtrless}(t)e^{i\omega_{\mathbf{q}\alpha}(t-t')}\right]. \tag{B.3}$$

The expression for $G_{\mathbf{k}\mu\nu}^>(t,t')$ is identical provided that $f_{\mathbf{k}\mu}^{el} \to f_{\mathbf{k}\mu}^{el}-1$, see (18a). Implementing the same simplifications leading to the SEPE we obtain the same equations as in Table 1 but with different phononic scattering terms (the electronic scattering terms $S^{GW}$ and $S^{FM}$ remain unchanged). In particular the scattering terms for the phononic occupations and coherences read [compare with Eqs. (88) and (89)]

$$S_{\mathbf{q}\alpha}^{ph-occ} = 2\pi \sum_{\mathbf{k}\mu\nu} \frac{|g_{\mathbf{q}\alpha,\mu\nu}^d(\mathbf{k})|^2}{2\omega_{\mathbf{q}\alpha}}\left\{\delta\left(\epsilon_{\mathbf{q}+\mathbf{k}\nu} - \epsilon_{\mathbf{k}\mu} - \omega_{\mathbf{q}\alpha}\right)\right.$$
$$\left.\times\left[\left(f_{\mathbf{q}+\mathbf{k}\nu}^{el} - 1\right)f_{\mathbf{k}\mu}^{el}\left(f_{\mathbf{q}\alpha}^{ph} + \text{Re}[\Theta_{\mathbf{q}\alpha}]\right) - f_{\mathbf{q}+\mathbf{k}\nu}^{el}\left(f_{\mathbf{k}\mu}^{el} - 1\right)\left(f_{\mathbf{q}\alpha}^{ph} + 1 + \text{Re}[\Theta_{\mathbf{q}\alpha}]\right)\right]\right\}, \tag{B.4}$$

$$S_{\mathbf{q}\alpha}^{ph-coh} = 2\pi \sum_{\mathbf{k}\mu\nu} \frac{|g_{\mathbf{q}\alpha,\mu\nu}^d(\mathbf{k})|^2}{2\omega_{\mathbf{q}\alpha}}$$
$$\times\left\{-\delta\left(\epsilon_{\mathbf{q}+\mathbf{k}\nu} - \epsilon_{\mathbf{k}\mu} - \omega_{\mathbf{q}\alpha}\right)\right.$$
$$\times\left[\left(f_{\mathbf{q}+\mathbf{k}\nu}^{el} - 1\right)f_{\mathbf{k}\mu}^{el}\left(f_{\mathbf{q}\alpha}^{ph} + \Theta_{\mathbf{q}\alpha}\right) - f_{\mathbf{q}+\mathbf{k}\nu}^{el}\left(f_{\mathbf{k}\mu}^{el} - 1\right)\left(f_{\mathbf{q}\alpha}^{ph} + 1 + \Theta_{\mathbf{q}\alpha}\right)\right]$$
$$+ \delta\left(\epsilon_{\mathbf{q}+\mathbf{k}\nu} - \epsilon_{\mathbf{k}\mu} + \omega_{\mathbf{q}\alpha}\right)$$
$$\left.\times\left[\left(f_{\mathbf{q}+\mathbf{k}\nu}^{el} - 1\right)f_{\mathbf{k}\mu}^{el}\left(f_{-\mathbf{q}\alpha}^{ph} + 1 + \Theta_{\mathbf{q}\alpha}\right) - f_{\mathbf{q}+\mathbf{k}\nu}^{el}\left(f_{\mathbf{k}\mu}^{el} - 1\right)\left(f_{-\mathbf{q}\alpha}^{ph} + \Theta_{\mathbf{q}\alpha}\right)\right]\right\}. \tag{B.5}$$

The equation of motion for the real and imaginary part of the coherences are identical to Eqs. (90), but the sign of $\Gamma_{\mathbf{q}\alpha}^{coh}$ is reversed.

The GKBA poses issues when employed to estimate polarization rates as well. In MGKBA the Fan-Migdal contribution to $-\Gamma_{\mathbf{k}\mu\mu}^{el,>}p_{\mathbf{k}\mu\nu}$, see Eqs. (71) and (73), is calculated with the po-

larization rate

$$\Gamma^{\text{el},>}_{\mathbf{k}\mu\mu} = \pi \sum_{\mathbf{q}\nu'\alpha} \frac{\left|g^{d}_{\mathbf{q}-\mathbf{k}\alpha,\mu\nu'}(\mathbf{k})\right|^{2}}{2\omega_{\mathbf{k}-\mathbf{q}\alpha}} \Big\{ \delta\big(\epsilon_{\mathbf{q}\nu'} - \epsilon_{\mathbf{k}\mu} + \omega_{\mathbf{k}-\mathbf{q}\alpha}\big)\big(1 - f^{\text{el}}_{\mathbf{q}\nu'}\big)\text{Re}\big[B^{>}_{\mathbf{k}-\mathbf{q}\alpha}\big]$$
$$+ \delta\big(\epsilon_{\mathbf{q}\nu'} - \epsilon_{\mathbf{k}\mu} - \omega_{\mathbf{k}-\mathbf{q}\alpha}\big)\big(1 - f^{\text{el}}_{\mathbf{q}\nu'}\big)\text{Re}\big[B^{<}_{\mathbf{k}-\mathbf{q}\alpha}\big]\Big\}. \qquad \text{(B.6)}$$

For the argument of the Dirac delta to vanish the index $\nu'$ must belong to the same "class" (conduction or valence) as the index $\mu$ and therefore the polarization rate is dominated by either conduction-conduction or valence-valence *e-ph* couplings. In GKBA the same term is replaced by $-\Gamma^{\text{el},>}_{\mathbf{k}\mu\nu}p_{\mathbf{k}\mu\nu}$ with

$$\Gamma^{\text{el},>}_{\mathbf{k}\mu\nu} = \pi \sum_{\mathbf{q}\nu'\alpha} \frac{\left|g^{d}_{\mathbf{q}-\mathbf{k}\alpha,\mu\nu'}(\mathbf{k})\right|^{2}}{2\omega_{\mathbf{k}-\mathbf{q}\alpha}} \Big\{ \delta\big(\epsilon_{\mathbf{q}\nu'} - \epsilon_{\mathbf{k}\nu} + \omega_{\mathbf{k}-\mathbf{q}\alpha}\big)\big(1 - f^{\text{el}}_{\mathbf{q}\nu'}\big)\text{Re}\big[B^{>}_{\mathbf{k}-\mathbf{q}\alpha}\big]$$
$$+ \delta\big(\epsilon_{\mathbf{q}\nu'} - \epsilon_{\mathbf{k}\nu} - \omega_{\mathbf{k}-\mathbf{q}\alpha}\big)\big(1 - f^{\text{el}}_{\mathbf{q}\nu'}\big)\text{Re}\big[B^{<}_{\mathbf{k}-\mathbf{q}\alpha}\big]\Big\}, \qquad \text{(B.7)}$$

i.e., the argument of the Dirac delta is calculated with $\epsilon_{\mathbf{k}\nu}$ instead of $\epsilon_{\mathbf{k}\mu}$. Using the same reasoning, we infer that the polarization rate is dominated by either conduction-valence or valence-conduction *e-ph* couplings, which is not to be expected. Moreover, there is no guarantee that the matrix $\Gamma^{\text{el},>}_{\mathbf{k}\mu\nu}$ is positive semi-definite for quasi-thermal distributions.

## C  H-theorem for coupled electron-phonon systems

In this appendix we prove that the total entropy of the coupled system of electrons and phonons, evolving in accordance to the Boltzmann equations, is a monotonically increasing function of time. This is a generalization of the *H-theorem* to multicomponent systems. In the context of the Boltzmann equations formulation the total entropy of the system is $S = S^{\text{el}} + S^{\text{ph}}$, where

$$S^{\text{el}} = -K_{\text{B}} \sum_{\mathbf{k}\mu} \Big[ f^{\text{el}}_{\mathbf{k}\mu} \ln f^{\text{el}}_{\mathbf{k}\mu} + (1 - f^{\text{el}}_{\mathbf{k}\mu})\ln(1 - f^{\text{el}}_{\mathbf{k}\mu}) \Big], \qquad \text{(C.1a)}$$

$$S^{\text{ph}} = -K_{\text{B}} \sum_{\mathbf{q}\alpha} \Big[ f^{\text{ph}}_{\mathbf{q}\alpha} \ln f^{\text{ph}}_{\mathbf{q}\alpha} - (1 + f^{\text{ph}}_{\mathbf{q}\alpha})\ln(1 + f^{\text{ph}}_{\mathbf{q}\alpha}) \Big], \qquad \text{(C.1b)}$$

and $K_{\text{B}}$ the Boltzmann constant. The H-theorem establishes that the quantity $H = -S/K_{\text{B}}$ has a non-positive time-derivative, i.e., $dH/dt \le 0$.

Let $H^{\text{el}} = -S^{\text{el}}/K_{\text{B}}$ and $H^{\text{ph}} = -S^{\text{ph}}/K_{\text{B}}$, so that $H = H^{\text{el}} + H^{\text{ph}}$. We have

$$\frac{dH^{\text{el}}}{dt} = \sum_{\mathbf{k}\mu} \frac{df^{\text{el}}_{\mathbf{k}\mu}}{dt} \ln \frac{f^{\text{el}}_{\mathbf{k}\mu}}{1 - f^{\text{el}}_{\mathbf{k}\mu}} = \sum_{\mathbf{k}\mu} (S^{\text{GW}}_{\mathbf{k}\mu\mu} + S^{\text{FM}}_{\mathbf{k}\mu\mu}) \ln \frac{f^{\text{el}}_{\mathbf{k}\mu}}{1 - f^{\text{el}}_{\mathbf{k}\mu}}. \qquad \text{(C.2)}$$

The GW scattering term is given in Eq. (69) and can be rewritten as

$$S^{\text{GW}}_{\mathbf{k}\mu\mu} = 2\pi \sum_{\mu'\nu\nu'} \sum_{\mathbf{k}'\mathbf{pp}'} \left|W_{\mathbf{k}\mu\mathbf{k}'\nu'\mathbf{p}\nu\mathbf{p}'\mu'}\right|^{2} \delta\big(\epsilon_{\mathbf{p}\nu} + \epsilon_{\mathbf{p}'\mu'} - \epsilon_{\mathbf{k}'\nu'} - \epsilon_{\mathbf{k}\mu}\big)\delta_{\mathbf{k}+\mathbf{k}',\mathbf{p}+\mathbf{p}'}$$
$$\times \Big[ \bar{f}^{\text{el}}_{\mathbf{k}'\nu'} \bar{f}^{\text{el}}_{\mathbf{k}\mu} f^{\text{el}}_{\mathbf{p}\nu} f^{\text{el}}_{\mathbf{p}'\mu'} - f^{\text{el}}_{\mathbf{k}'\nu'} f^{\text{el}}_{\mathbf{k}\mu} \bar{f}^{\text{el}}_{\mathbf{p}\nu} \bar{f}^{\text{el}}_{\mathbf{p}'\mu'} \Big], \qquad \text{(C.3)}$$

where we define $\bar{f}^{\text{el}}_{\mathbf{k}\mu} = 1 - f^{\text{el}}_{\mathbf{k}\mu} \ge 0$. To recognize the mathematical structure of this equation we find it convenient to introduce the superindices $i = \mathbf{k}\mu$, $j = \mathbf{k}'\nu'$, $m = \mathbf{p}\nu$ and $n = \mathbf{p}'\mu'$.

Then, the GW contribution to (C.2) reads

$$\frac{dH^{\text{el}}}{dt}\bigg|_{\text{GW}} = \sum_{ijmn} \Gamma_{ijmn}\Big[\bar{f}_i^{\text{el}}\bar{f}_j^{\text{el}}f_m^{\text{el}}f_n^{\text{el}} - f_i^{\text{el}}f_j^{\text{el}}\bar{f}_m^{\text{el}}\bar{f}_n^{\text{el}}\Big]\ln\frac{f_i^{\text{el}}}{\bar{f}_i^{\text{el}}}, \tag{C.4}$$

with

$$\Gamma_{ijmn} = 2\pi\big|W_{ijmn}\big|^2\delta\big(\epsilon_i + \epsilon_j - \epsilon_m - \epsilon_n\big)\delta_{\mathbf{k}_i+\mathbf{k}_j,\mathbf{k}_m+\mathbf{k}_n} = \Gamma_{jinm} = \Gamma_{nmji} \geq 0, \tag{C.5}$$

and $\mathbf{k}_i$ the momentum carried by the superindex $i$, $\mathbf{k}_j$ the momentum carried by the superindex $j$, etc. The term in the square bracket of Eq. (C.4) is symmetric under the exchange $i \leftrightarrow j$ and $m \leftrightarrow n$, whereas it is antisymmetric under the exchange $i \leftrightarrow n$ and $j \leftrightarrow m$. Therefore we can rewrite (C.4) as

$$\frac{dH^{\text{el}}}{dt}\bigg|_{\text{GW}} = \sum_{ijmn} \frac{\Gamma_{ijmn}}{4}\Big[\bar{f}_i^{\text{el}}\bar{f}_j^{\text{el}}f_m^{\text{el}}f_n^{\text{el}} - f_i^{\text{el}}f_j^{\text{el}}\bar{f}_m^{\text{el}}\bar{f}_n^{\text{el}}\Big]\left[\ln\frac{f_i^{\text{el}}}{\bar{f}_i^{\text{el}}} + \ln\frac{f_j^{\text{el}}}{\bar{f}_j^{\text{el}}} - \ln\frac{f_m^{\text{el}}}{\bar{f}_m^{\text{el}}} - \ln\frac{f_n^{\text{el}}}{\bar{f}_n^{\text{el}}}\right]$$

$$= \sum_{ijmn} \frac{\Gamma_{ijmn}}{4}\bar{f}_i^{\text{el}}\bar{f}_j^{\text{el}}f_m^{\text{el}}f_n^{\text{el}}\left[1 - \frac{f_i^{\text{el}}f_j^{\text{el}}\bar{f}_m^{\text{el}}\bar{f}_n^{\text{el}}}{\bar{f}_i^{\text{el}}\bar{f}_j^{\text{el}}f_m^{\text{el}}f_n^{\text{el}}}\right]\ln\frac{f_i^{\text{el}}f_j^{\text{el}}\bar{f}_m^{\text{el}}\bar{f}_n^{\text{el}}}{\bar{f}_i^{\text{el}}\bar{f}_j^{\text{el}}f_m^{\text{el}}f_n^{\text{el}}}. \tag{C.6}$$

Taking into account that $\Gamma_{ijmn}\bar{f}_i^{\text{el}}\bar{f}_j^{\text{el}}f_m^{\text{el}}f_n^{\text{el}} \geq 0$ and that the function $(1-x)\ln x \leq 0$ for all $x > 0$ we conclude that

$$\frac{dH^{\text{el}}}{dt}\bigg|_{\text{GW}} \leq 0. \tag{C.7}$$

Let us now come to the Fan-Migdal contribution. The Fan-Migdal scattering term is given in Eq. (70) and can be rewritten as

$$S_{\mathbf{k}\mu\mu}^{\text{FM}} = -2\pi\sum_{\mathbf{q}\alpha\mathbf{k}'\nu}\frac{\big|g_{-\mathbf{q}\alpha,\mu\nu}(\mathbf{k})\big|^2}{2\omega_{\mathbf{q}\alpha}}\delta\big(\epsilon_{\mathbf{k}'\nu} - \epsilon_{\mathbf{k}\mu} + \omega_{\mathbf{q}\alpha}\big)\delta_{\mathbf{q},\mathbf{k}-\mathbf{k}'}\Big[\bar{f}_{\mathbf{k}'\nu}^{\text{el}}f_{\mathbf{k}\mu}^{\text{el}}\bar{f}_{\mathbf{q}\alpha}^{\text{ph}} - f_{\mathbf{k}'\nu}^{\text{el}}\bar{f}_{\mathbf{k}\mu}^{\text{el}}f_{\mathbf{q}\alpha}^{\text{ph}}\Big]$$

$$-2\pi\sum_{\mathbf{q}\alpha\mathbf{k}'\nu}\frac{\big|g_{\mathbf{q}\alpha,\mu\nu}(\mathbf{k})\big|^2}{2\omega_{\mathbf{q}\alpha}}\delta\big(\epsilon_{\mathbf{k}'\nu} - \epsilon_{\mathbf{k}\mu} - \omega_{\mathbf{q}\alpha}\big)\delta_{\mathbf{q},\mathbf{k}'-\mathbf{k}}\Big[\bar{f}_{\mathbf{k}'\nu}^{\text{el}}f_{\mathbf{k}\mu}^{\text{el}}f_{\mathbf{q}\alpha}^{\text{ph}} - f_{\mathbf{k}'\nu}^{\text{el}}\bar{f}_{\mathbf{k}\mu}^{\text{el}}\bar{f}_{\mathbf{q}\alpha}^{\text{ph}}\Big], \tag{C.8}$$

where we define $\bar{f}_{\mathbf{q}\alpha}^{\text{ph}} = f_{\mathbf{q}\alpha}^{\text{ph}} + 1 \geq 0$. Let us introduce the quantity

$$\Gamma_{\mathbf{k}\mu\mathbf{k}'\nu\mathbf{q}\alpha} \equiv 2\pi\frac{\big|g_{-\mathbf{q}\alpha,\mu\nu}(\mathbf{k})\big|^2}{2\omega_{\mathbf{q}\alpha}}\delta\big(\epsilon_{\mathbf{k}'\nu} - \epsilon_{\mathbf{k}\mu} + \omega_{\mathbf{q}\alpha}\big)\delta_{\mathbf{q},\mathbf{k}-\mathbf{k}'} \geq 0. \tag{C.9}$$

Using the property in Eq. (7) we can rewrite Eq. (C.8) in the following compact form

$$S_{\mathbf{k}\mu\mu}^{\text{FM}} = -\sum_{\mathbf{q}\alpha\mathbf{k}'\nu}\Gamma_{\mathbf{k}\mu\mathbf{k}'\nu\mathbf{q}\alpha}\Big[\bar{f}_{\mathbf{k}'\nu}^{\text{el}}f_{\mathbf{k}\mu}^{\text{el}}\bar{f}_{\mathbf{q}\alpha}^{\text{ph}} - f_{\mathbf{k}'\nu}^{\text{el}}\bar{f}_{\mathbf{k}\mu}^{\text{el}}f_{\mathbf{q}\alpha}^{\text{ph}}\Big] - \sum_{\mathbf{q}\alpha\mathbf{k}'\nu}\Gamma_{\mathbf{k}'\nu\mathbf{k}\mu\mathbf{q}\alpha}\Big[\bar{f}_{\mathbf{k}'\nu}^{\text{el}}f_{\mathbf{k}\mu}^{\text{el}}f_{\mathbf{q}\alpha}^{\text{ph}} - f_{\mathbf{k}'\nu}^{\text{el}}\bar{f}_{\mathbf{k}\mu}^{\text{el}}\bar{f}_{\mathbf{q}\alpha}^{\text{ph}}\Big]. \tag{C.10}$$

Again, we highlight the mathematical structure by introducing the superindices $i = \mathbf{k}\mu$, $j = \mathbf{k}'\nu'$, and $q = \mathbf{q}\alpha$. Let us define

$$R_{ijq} \equiv \Gamma_{ijq}\big[f_i^{\text{el}}\bar{f}_j^{\text{el}}\bar{f}_q^{\text{ph}} - \bar{f}_i^{\text{el}}f_j^{\text{el}}f_q^{\text{ph}}\big] = -\Gamma_{ijq}\bar{f}_i^{\text{el}}f_j^{\text{el}}f_q^{\text{ph}}\left[1 - \frac{f_i^{\text{el}}\bar{f}_j^{\text{el}}\bar{f}_q^{\text{ph}}}{\bar{f}_i^{\text{el}}f_j^{\text{el}}f_q^{\text{ph}}}\right]. \tag{C.11}$$

Then $S^{\text{FM}}_{\mathbf{k}\mu\mu} = -\sum_{jq}(R_{ijq} - R_{jiq})$ and the Fan-Migdal contribution to Eq. (C.2) reads

$$
\begin{aligned}
\left.\frac{d\text{H}^{\text{el}}}{dt}\right|_{\text{FM}} &= -\sum_{ijq}(R_{ijq} - R_{jiq})\ln\frac{f_i^{\text{el}}}{\bar{f}_i^{\text{el}}} = -\sum_{ijq}R_{ijq}\left[\ln\frac{f_i^{\text{el}}}{\bar{f}_i^{\text{el}}} - \ln\frac{f_j^{\text{el}}}{\bar{f}_j^{\text{el}}}\right] \\
&= -\sum_{ijq}R_{ijq}\left[\ln\frac{f_i^{\text{el}}\bar{f}_j^{\text{el}}\bar{f}_q^{\text{ph}}}{\bar{f}_i^{\text{el}}f_j^{\text{el}}f_q^{\text{ph}}} - \ln\frac{\bar{f}_q^{\text{ph}}}{f_q^{\text{ph}}}\right].
\end{aligned}
\tag{C.12}
$$

Next we calculate $d\text{H}^{\text{ph}}/dt$. Using Eq. (C.1b) we find

$$
\frac{d\text{H}^{\text{ph}}}{dt} = \sum_{\mathbf{q}\alpha}\frac{df_{\mathbf{q}\alpha}^{\text{ph}}}{dt}\ln\frac{f_{\mathbf{q}\alpha}^{\text{ph}}}{\bar{f}_{\mathbf{q}\alpha}^{\text{ph}}} = \sum_{\mathbf{q}\alpha}S^{\text{ph-occ}}_{\mathbf{q}\alpha\alpha}\ln\frac{f_{\mathbf{q}\alpha}^{\text{ph}}}{\bar{f}_{\mathbf{q}\alpha}^{\text{ph}}}.
\tag{C.13}
$$

The phononic scattering term is given by Eq. (88) with $\Theta_{\mathbf{q}\alpha} = 0$, and can be rewritten as

$$
\begin{aligned}
S^{\text{ph-occ}}_{\mathbf{q}\alpha\alpha}(t) &= -2\pi\sum_{\mathbf{k}\mu\mathbf{k}'\nu}\frac{|g_{\mathbf{q}\alpha,\mu\nu}(\mathbf{k})|^2}{2\omega_{\mathbf{q}\alpha}}\delta(\epsilon_{\mathbf{k}'\nu} - \epsilon_{\mathbf{k}\mu} - \omega_{\mathbf{q}\alpha})\delta_{\mathbf{q},\mathbf{k}'-\mathbf{k}}\left[\bar{f}_{\mathbf{k}'\nu}^{\text{el}}f_{\mathbf{k}\mu}^{\text{el}}f_{\mathbf{q}\alpha}^{\text{ph}} - f_{\mathbf{k}'\nu}^{\text{el}}\bar{f}_{\mathbf{k}\mu}^{\text{el}}\bar{f}_{\mathbf{q}\alpha}^{\text{ph}}\right] \\
&= -\sum_{\mathbf{k}\mu\mathbf{k}'\nu}\Gamma_{\mathbf{k}'\nu\mathbf{k}\mu\mathbf{q}\alpha}\left[\bar{f}_{\mathbf{k}'\nu}^{\text{el}}f_{\mathbf{k}\mu}^{\text{el}}f_{\mathbf{q}\alpha}^{\text{ph}} - f_{\mathbf{k}'\nu}^{\text{el}}\bar{f}_{\mathbf{k}\mu}^{\text{el}}\bar{f}_{\mathbf{q}\alpha}^{\text{ph}}\right] = \sum_{ij}R_{jiq}.
\end{aligned}
\tag{C.14}
$$

Inserting this result in Eq. (C.13) and using superindices we obtain the following compact expression

$$
\frac{d\text{H}^{\text{ph}}}{dt} = \sum_{ijq}R_{ijq}\ln\frac{f_q^{\text{ph}}}{\bar{f}_q^{\text{ph}}},
\tag{C.15}
$$

which is the negative of the last term in Eq. (C.12). We conclude that

$$
\begin{aligned}
\left.\frac{d\text{H}^{\text{el}}}{dt}\right|_{\text{FM}} + \frac{d\text{H}^{\text{ph}}}{dt} &= -\sum_{ijq}R_{ijq}\ln\frac{f_i^{\text{el}}\bar{f}_j^{\text{el}}\bar{f}_q^{\text{ph}}}{\bar{f}_i^{\text{el}}f_j^{\text{el}}f_q^{\text{ph}}} \\
&= \sum_{ijq}\Gamma_{ijq}\bar{f}_i^{\text{el}}f_j^{\text{el}}f_q^{\text{ph}}\left[1 - \frac{f_i^{\text{el}}\bar{f}_j^{\text{el}}\bar{f}_q^{\text{ph}}}{\bar{f}_i^{\text{el}}f_j^{\text{el}}f_q^{\text{ph}}}\right]\ln\frac{f_i^{\text{el}}\bar{f}_j^{\text{el}}\bar{f}_q^{\text{ph}}}{\bar{f}_i^{\text{el}}f_j^{\text{el}}f_q^{\text{ph}}}.
\end{aligned}
\tag{C.16}
$$

Taking into account that $\Gamma_{ijq}\bar{f}_i^{\text{el}}f_j^{\text{el}}f_q^{\text{ph}} \geq 0$ and that the function $(1-x)\ln x \leq 0$ for all $x > 0$, we infer that the r.h.s. of Eq. (C.16) is non-positive. This result together with Eq. (C.7) concludes the proof of the H-theorem for coupled electron-phonon systems.

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
