# Peer review of "Semiconductor Electron-Phonon Equations: a Rung Above Boltzmann in the Many-Body Ladder"

_SciPost Physics, doi:SciPost Phys. 16, 073 (2024)_

## Round 1 · Referee Report · Dino Novko (Referee 1) · 2023-12-21

Strengths
-
The paper is clearly written with a very logical and reasonable flow of derivations and discussions.
-
With increasing interest into ultrafast dynamics and interactions on the ultrashort time scale, the paper provides a necessary and numerically feasible methodology that will definitely unveil some new insights into coherent phonon dynamics and related phenomena. Therefore, the paper is very timely.
Weaknesses
Report
Here are some comments, that authors can optionally consider:
1.) In the introduction when you first mention Boltzmann equations, some appropriate citation would be desirable, considering that you introduced a citation for SBE.
2.) Considering that the present model is framed within the BO and harmonic approximation, how useful would this be for soft phonon dynamics, charge density waves, and phase transitions? Can it be extended to cover also phonon energy renormalizations?
Of course, anharmonicitiy is crucial, but CDW and soft phonons that come from the electron-phonon coupling can be tackled with higher orders of el-phon (via phonon self energy):
https://journals.aps.org/prl/abstract/10.1103/PhysRevLett.51.138
also within the (quasi)harmonic limit..
Such descriptions of soft phonon dynamics and phase transitions in time domain has become a pressing matter:
https://arxiv.org/abs/2203.11880
But most of the descriptions are still semi-classical.
Going over the CDW transition would be out of reach for BO harmonic approx., but maybe treating the CDW amplitude phonon mode as a coherent oscillation q=0, and inspecting the CDW system below T<T_CDW.
3.) From the perspective of other previous work (what you called model Hamiltonians, but also some ab initio DFT works), the resoning behind Eq. 5 is not completely clear to me. In some interprations \Pi is already in K.. Or other way around.. I think later you call this quasi-pasticle Q..
What would be the physical meaning of the frequency of a phonon if you would leave out \Pi (i.e., what is a bare frequency here)? Only ion-ion contribution? When g=0?
It is an old problem:
https://iopscience.iop.org/article/10.1070/PU1975v018n03ABEH001953/meta
And maybe should be discussed more (or maybe not).
4.) From Eq. 16 it turns out that external perturbation induces coherent q=0 phonon motion, that do not brake the original periodicity of the unit cell.. However, is it possible to capture the effect where this coherent phonon transfers the energy to other q>0 via electron mediated phonon-phonon coupling?
Somehow a coupling between nuclear motion Eq. 16 and electron-phonon system described with Eqs. 13a and 13b should allow for that?
5.) In Figure 1 the curly phonon line is not defined, also, triangle vertex are not defined as well as double straight line for G. Maybe should be extended, so that reader does not need to go into your recent PRX.
6.) What are the benefits of SEPE and MGKBA equations derived in this paper compared to:
https://journals.aps.org/rmp/abstract/10.1103/RevModPhys.86.779
https://journals.aps.org/prb/abstract/10.1103/PhysRevB.91.045128
https://journals.aps.org/prb/abstract/10.1103/PhysRevB.93.094509
7.) Regarding the discussion after Eq. 78, this important difference that you mention is not clear enough here.. The equation of motion for U_0 in "model Hamiltonians" (as you refer to it even though the present Hamiltonian is also a model Hamiltonian) differs from Eq. 78 I guess because of different definition of starting bare \omega_0^2.. Your \omega_0 is different from the one in this model Hamiltonian, where you separate elastic tensor from phonon self energy due to el-ion interaction.
Double renormalization then comes if one does not think carefully of the starting point for \omega_0.
8.) What is the reason you leave out the time dependency of Q_qp for the phonon part of equations of motion? For electronic quasi-particle energies you consider some form of time evolution, at least on the LHS part of the SEPE equation (i.e., in the case of electronic polarization).
9.) Regarding the dicussion that follows the Eq. 90, I wonder what happens with coherence rate \Gamma_^{coh} if your distribution of carriers is non-thermal. Even if you start with quasi-thermalized f_k, the particular phase space and coupling constants can result in non-thermal distributions. What is the meaning of negative \Gamma^{coh} in that case?
10.) Regarding statement that the BE do not account for the interaction of field and electrons, I would like to point out that there are versions of BE where field is included, so in principle BE have the driving field. But usually it is not included for simplicity.
Just as an example:
https://journals.aps.org/prb/abstract/10.1103/PhysRevB.65.214303
https://journals.aps.org/prb/abstract/10.1103/PhysRevB.74.024301
https://journals.aps.org/prb/abstract/10.1103/PhysRevB.96.174439
These are just listed as an example, do not need to cite them.
11.) In the conclusion you mention that only W and g are necessary to run SEPE. I would say that you also need electron energies and phonon energies, which is big difference. For instance, in some spectral representation of equations of motion like BE, it is possible to have only W and g, but in addition you don't need energies, just some DOS. Here you derive the equations so that you need energies in band and momentum resolution (with anisotropy of the phase space), which can be quite a numerical obstacle.
Requested changes
1.) In Eq. 4b, is it q=0 or -q in the last term? There is 0 in g at the moment.
2.) Between Eq. 51 and 52, the sentence should be maybe without "with" at the end.
3.) In conclusion, when you discuss the ph-ph implementation, you should write "which are expected TO play a role in the thermalization of the lattice"
Author: Gianluca Stefanucci on 2024-01-12 [id 4242]
(in reply to Report 1 by Dino Novko on 2023-12-21)
We thank the Referee for the careful reading and for providing insightful suggestions and valuable comments. We provide below a point-by-point reply.
1) It is not easy to assign citations to the Boltzmann equations without doing injustice to many authors who contributed with fundamental mathematical results and applications. We have decided to cite two classic books (the Kadanoff-Baym and the Landau-Pitaevskii-Lifshitz) where the reader can find an introduction to the topic along with many insightful discussions. This choice reflects our personal formation and is by no means intended to be exhaustive.
2) We thank the Referee for this interesting remark. Although the MGKBA form of the self-energy proposed by Varma and Simons can be derived, we see that it leads to integro-differential equations for the density matrices due to the presence of internal vertices. More generally, any many-body approximation beyond the one considered in this work is responsible for changing the time scaling from linear to quadratic or more (or alternatively for increasing the number of unknown quantities to propagate).
About the possibility of studying CDW's. The correlation function of the local lattice displacements, as defined in Eq.(9) of the work by Petrović et al, can be extracted from the full phononic density matrix (or equivalently from the phononic occupations and coherences) at q=\pi. However, these correlation functions do not fully characterize a CDW system. The CDW order parameter is a density matrix nondiagonal in k-space, which is set to zero in our equations as we work under the hypothesis that the lattice periodicity is not broken by the external field. Commensurate distortion of the lattice can be handled as described in our reply to point 4).
3) The bare electron-phonon frequencies are given by the square root of the eigenvalues of the elastic tensor K. This tensor is the sum of the only ion-ion contribution and the Debey-Waller coupling multiplied by the equilibrium density, see Eq.(14) in our recent PRX. The eigenvalues of K provide the "spring constants" acting on the nuclei when keeping the electrons frozen in equilibrium (as such they are not physical and they may even be negative). The fact that a Hamiltonian written in terms of dressed phonon frequencies implies the exclusion of the zero-frequency RPA diagrams from the phononic self-energy is an old but still not so spread message. In fact, this aspect becomes especially critical in the time domain (Kadanoff-Baym equations (KBE)) as the zero-frequency contribution to \Pi(t,t') cannot be isolated. In our work we have shown how to deal with this delicate issue through the MGKBA. We observe that the same aspect becomes critical also in numerically exact solutions of model Hamiltonians since, by construction, they are equivalent to summing all diagrams, and these include those of the RPA series (whose w=0 contribution should instead be subtracted).
4) No, it is not possible. If the external driving does not break the lattice periodicity the invariance of the system under discrete translations is preserved during the entire time evolution and consequently Uq=0 for nonvanishing q. A possible way to study commensurate distortions of the lattice consists in using supercells and in breaking the original symmetry with fields having the periodicity of the supercell lattice. In the revised version we have added a comment on this point at the end of Section 2.
5) In the figure caption we now explain the meaning of the various graphical quantities
6) In the papers mentioned by the Referee the authors solve the KBE with a DMFT self-energy. In general the KBE are superior to the SEPE, SBE and BE since they do not rely on the MGKBA and Markovian approximation. As pointed out in the introduction, the motivation for deriving simplified methods is exclusively related to the numerical cost of the KBE. After about 20 years since the first implementation of the KBE the applications are still limited to rather simple model Hamiltonians. The original contribution given in the papers mentioned by the Referee is in the self-energy (DMFT approximation) rather than in the simplification of the original KBE equations.
7) We must disagree with the statements contained in this point. The Hamiltonian in Eq.(4) is the ab initio Hamiltonian written in terms of second-quantized operators expanded over some suitable orthonormal basis, hence it is not a model Hamiltonian. The \omega_0 in Eq.(79) is the physical phonon frequency at q=0 defined in Eq.(5) [the key step in the derivation is Eq.(78)] and used in model Hamiltonians. Thus the comparison between Eq.(79) and the equation of motion of model Hamiltonians makes perfect sense. We stress again that model Hamiltonians are written in terms of physical phonon frequencies, and hence already account for a piece of the phononic self-energy. The analogue in the purely electronic case would be to write the Hamiltonian in terms of the Kohn-Sham (KS) energies and then use the self-energy Sigma instead of the difference (\Sigma-v_Hxc) in the Dyson equation. In the revised manuscript we emphasize that the root cause of the problem lies in the unjustified replacement of Eq.(4b) with a set of harmonic oscillators.
8) The origin of the time-dependence in Q_qp stems from the response function evaluated with time-dependent electronic occupations, see Eqs.(29) and (50). This leads to a time-dependent renormalization of the phonon frequencies, an effect that may be important in some applications. We have here decided to ignore such renormalization as it would require the explicit calculation of Eq.(50) at every time-step. However, we agree with the Referee that this numerical complication does not represent an obstacle in the derivation of the SEPE, the only change being that \w -> \w(t) in Eq.(90). In the revised manuscript we have added a brief comment on this aspect at the end of Section 5.
9) This is not an easy question to answer since the phononic coherence is not associated to a hermitian operator. In fact, the equation of motion for \Theta does not have the classic form with an "emission" contribution and an "absorption" contribution. Although the rigorous mathematical study of the SEPE is not straightforward, on physical ground we expect that the effect of the coherences on the electronic occupation is subdominant with respect to the effect of the phononic occupations, see Eq.(88). Therefore the electronic occupations will approach a thermal distribution in the long run, implying that \Gamma^{coh}(t) will attain a positive value for t->infty. If one starts with a highly non-thermal distribution of carriers then the coherences are expected to initially grow and subsequently approach zero exponentially.
10) We agree with the Referee that our statement is not precise. Historically the BE contains the coupling between a longitudinal electric field E=-\grad V and the k-derivative of the occupations (in this form the BE becomes a system of partial differential equations). As we are here interested in transverse fields (laser fields) the semiconductor Bloch equations are more appropriate to describe the light-matter coupling. In the revised version we make our statement more precise.
11) We again agree with the Referee that our statement is not precise. In the revised version we clarify that in addition to W and g one also needs the electronic and phononic dispersions. However, this is not a numerical obstacle; electronic and phononic dispersions are routinely calculated for a vast class of materials. For quantitative and material specific predictions a knowledge of these dispersions is an unavoidable input.
Requested changes
1) Eq.(4b) is correct, see Eq.(D10) in our PRX and consider that in equilibrium only the q=0 term survives. The confusion may have arisen from the fact that we have defined g at finite q shortly thereafter. We have improved the discussion in the revised version.
2) We have corrected the typo
3) We have corrected the typo

Claudio Attaccalite on 2023-11-12 [id 4108]
Dear authors
Your manuscript will soon be reviewed by the referees, in the meantime I would like to ask you some questions about your work, which you can answer in your reply to the referees.
1) In your manuscript you use the MGKBA to derive SBE equations because the standard GKBA leads to non-physical results. However, in the literature there is a discussion on how to use the GKBA and in particular a possible linear combination of MGKBA or GKBA, see for example page 289-290 of the book "Semiconductor Optics and Transport Phenomena" by Wilfried Schäfer, Martin Wegener were additional conditions are derived in such a way that they "coincide with the exact results of the density matrix. "Can the authors comment on what is lost or changed by the MGKBA treatment of electrons compared to the standard GKBA?
2) On page 20 the authors cite a paper by Malic et al as an example of SBE in which phonon occupancies satisfy their own equation, I think this kind of equation is much older and also the application of the GKBA to the phonon self-energy, see Phys. Rev. B 52, 5624 (1995). I think the authors should cite this work.
3) Since in the SEBE formalism there is also the equation for the nuclear displacement, I wonder if this approach might be able to describe trapped excitons or light induced charge density waves?
Kind regards Claudio Attaccalite
Anonymous on 2024-01-12 [id 4243]
(in reply to Claudio Attaccalite on 2023-11-12 [id 4108])1) We thank the Editor for bringing our attention to a discussion in the book "Semiconductor Optics and Transport Phenomena" that we were not aware of. Schäfer and Wegener do indeed notice that the interchange of the retarded and advanced Green's functions leads to another legitimate solution of the collisionless approximation, and that in this case memory effects "vanish completely" (actually this is not entirely true for self-energies beyond the ones considered in our paper). This interchange is de facto equivalent to our MGKBA. Schäfer and Wegener also point out that any linear combination of GKBA and MGKBA is a solution of the collisionless approximation and "optimize" the combination for the off-diagonal Green's function in the low density limit. This is less relevant for us since we have discarded the off-diagonal matrix elements of the Green's functions in the evaluation of the scattering terms. In fact, any attempt to include the off-diagonal matrix elements (including the optimized one by Schäfer and Wegener) gives rise to an equation of motion that either violates the positivity of the density matrix or the trace preserving property. Although the MGKBA for the electronic case is contemplated in the book by Schäfer and Wegener, the analogue ansatz for phonons is missing and the consequences are not elaborated. In the revised version we cite the book by Schäfer and Wegener when we introduce the electronic MGKBA.
2) We agree with the Editor. Although the kinetic equation for the phononic occupation is derived for a special one-dimensional model the basic ideas are already contained in the work by Meden et al. In the revised version we cite this paper.
3) The SEPE have been derived assuming that the equilibrium lattice periodicity is not broken by the external field. As pointed out in our reply to point 4) of Referee 1 and to point 6) of Referee 2, the SEPE can easily be extended to deal with commensurate breaking of the equilibrium lattice periodicity. Thus trapped excitons or light induced charge density waves can be described provided that the new periodicity is an integer multiple of the equilibrium one. Of course, the greater the size of the supercell, the higher the numerical effort required.

---

## Round 1 · Referee Report · Claudio Verdozzi (Referee 2) · 2023-12-22

Strengths
1) This work introduces a new approach (SEPE) to the optical response of semiconductors
2) The approach provide an unified perspective of methods like the Bethe Salpether equation, the Boltzmann equation, the Bloch equations.
3) At the same time, it offers an equal footing account of the coherent and incoherent dynamics of phonons, and the influence of these (via renormalization of the quasi-particle energies) on the electrons.
Weaknesses
Report
The paper is well written. While there is plenty of formal derivations, the author adopt all throughout a pedagogical pace, which helps considerably in following the logical unfolding of the process to arrive at the SEPE. The review of the NEGF, (M)GKBA), etc techniques is very informative, and there is a terse formulation of the conceptual progression through a series of approximations and simplifications (S1-S4 in the paper) from general double-time NEGF, to NEGF-(M)GKBA, a and then SEPE (clearly and conveniently summarised in a table) and eventually Boltzmann equations. A great merit of the approach (and the paper) is that, at all stages of the simplification process, there a continuity of connection that traces back to the first principle KBE, so for example quantities like scattering rates can be rigorously connected back to many-body perturbation theory and full double-time Green’s functions. Furthermore, it is also very interesting that it is eventually the MKGBA and not GKBA that permits to proceed towards the SEPE and then the BE. Different limiting cases of the SEPE are also considered in the paper (e.g. the steady state regime), and the extensions foreseen by the authors for future work point to important developmental directions.
For all these reasons, I recommend publication in Scipost.
Requested changes
1) The discussion and the theoretical formulation in the paper relies at times on the general framework introduced in Ref [1]. The authors could review and summarize more explicitly and in some more detail (maybe in an appendix ?), the contents and main outcomes of [1], to facilitate the reader to better establish the connection with those conceptual steps taken in this paper that rely on [1].
2 ) the paper clearly describes what is incorporated in the SEPE. It would be interesting if a short discussion (maybe in the outlook) is also given of what is “left out” with the SEPE compared to “higher” rungs in the ladder.
3) The simplifications and approximations made to arrive at the SEPE seem plausible and are well motivated. However, to enhance the notion of how the approximations introduced to reach the SEPE perform, a concrete worked out example would be helpful, where SEPE and benchmarks results are compared. On the other hand, I find it quite reasonable that the authors have preferred to lay down first only the formal apparatus, but in a complete way and all in one place. So I leave for them to consider if in the revised manuscript such suggested concrete example would be included or not.
4) Minor typos: i) “nucleous” instead of “nucleus”, p.3, ii) “the second line of Eq. (4b)”, there is no second line in 4b, p.4, “is —> “are”, p.6 iii) simplifies to with —> simplifies to, p.10 iv) guarantess—> guarantee, p14 v) Please check the references , since in some cases capital letters in name have been demoted to lowercase, or the name of chemical compounds (e.g. mose2) is incorrectly rendered.
Questions that the authors can optionally consider:
5) Do the authors expect that the quality of the SEPE description would depend strongly with the size of the gap, once the system is gapped? In other words, what about doped semiconductors ?
6) In Eq.9, it is assumed that the external field has the lattice periodicity. What is to be expected if the driving field breaks such periodicity?
7) Have the authors considered the possibility (if viable) of extending the SEPE to also include quantised external electromagnetic fields ?
8) Can the approach be used for open gapped system, like in e.g. quantum transport setups with reservoirs ?
9) In Sect.7.5, the authors discuss the steady state for the SEPE, which involves in a natural way the notion of temperature, and its increase due to the injection of energy from the external field. Starting from a T=0 initial equilibrium state (i.e, an isolated semiconductor) and exposing it to a perturbation of finite duration, the steady state limit could be reached via the time evolution of the SEPE, where final phonon and electron temperatures might be reached within different time scales, and across a number of pre-thermalization stages. Could one encounter (and avoid) numerically the potential issue that the steady state determined numerically by the SEPE could be one of these intermediate stationary state, rather than the actual final steady state?
Author: Gianluca Stefanucci on 2024-01-12 [id 4241]
(in reply to Report 2 by Claudio Verdozzi on 2023-12-22)
We thank the Referee for the careful reading and for providing insightful suggestions and valuable comments. We provide below a point-by-point reply.
1) The general framework introduced in Ref [1] is summarized in Section 2 and 3. In fact, the only results of Ref.[1] that are needed in this work are the ab initio Hamiltonian, see Eqs.(4), and the ab initio KBE, see Eqs.(13) and (16). We have stated this point in the last paragraph of the introduction.
2) In the concluding remarks of the revised version we have summarized the approximations and simplifications implemented to derive the SEPE. This should clarify what is “left out”.
3) Our motivation for publishing a paper presenting exclusively the theoretical aspects of the formalism is twofold. First, we can show all derivations and discuss many subtle points related to the underlying simplifications and approximations of the SEPE, and hence of the widespread SBE and BE, without making the paper too long. Second, we believe that the implementation of the SEPE and their application to a physical problem is a contribution on its own.
4) We have corrected all typos
5) It is not easy to provide a quantitative answer to this question. The point is that the explicit form of the scattering terms before making the Markov approximation contains exponential integrals that, if not properly cured, initiate a non trivial dynamics even without external fields. The contributions responsible for initiating such artificial dynamics oscillate in time at a frequency of the order of the gap, and are therefore "small" on physical ground. When making the Markov approximation they become zero for arbitrary small gaps. We therefore expect that whenever non-Markovian effects are negligible the size of the gap does not affect the quality of the SEPE. This is a rather technical point and it is not a feature of the SEPE, as the same reasoning apply to the derivation of the Boltzmann equations.
6) The most general symmetry breaking makes the density matrices depend on two momenta rather than one. Although this complication can be mathematically handled, the scaling with the system size increases and the resulting equations become less appealing. However, if one is interested in commensurate symmetry breaking then it is enough to expand the fields, see Eqs. (1) and (2), in basis functions having the periodicity of the supercell lattice. In the revised version we have added a comment on this point at the end of Section 2.
7) This is another interesting observation. In fact, the (M)GKBA for phonons equally applies to photons. In the revised version we have added a comment in the concluding remarks.
8) Like the SBE and BE also the SEPE are expected to be accurate only for extended systems. In finite systems like molecules the simplification (S1) of diagonal density matrices and retarded GF is not justified. Therefore the extension to quantum transport setup is not straightforward.
9) As discussed in Section 7.5 the steady state is characterized by Fermi and Bose distribution at the same temperature. If an intermediate stationary state is characterized by distributions very close to the steady state ones then it is hard to say whether we have an intermediate or a final state. In the revised version we have added an Appendix proving the H-theorem for coupled systems of electrons and phonons described by the BE [to the best of our knowledge this is another original result], which is the long-time limit of the SEPE. Plotting the total entropy versus time could provide insights on the nature of the state.

---

## Round 2 · Referee Report · Dino Novko (Referee 1) · 2024-1-30

Report

I thank the authors for providing a detailed explanations on my comments and questions. I fully support the publication of the new and improved version of the manuscript.

---

## Round 2 · Referee Report · Claudio Verdozzi (Referee 2) · 2024-2-26

Strengths

1) This work introduces a new approach (SEPE) to the optical response of semiconductors 2) The approach provide an unified perspective of methods like the Bethe Salpether equation, the Boltzmann equation, the Bloch equations. 3) At the same time, it offers an equal footing account of the coherent and incoherent dynamics of phonons, and the influence of these (via renormalization of the quasi-particle energies) on the electrons. 4) A proof of the Boltzmann's H-theorem is given

Weaknesses

No particular weaknesses

Report

Following the indications of the editor and the referees,
in the revised paper the authors have introduced a number of changes which
have further improved the overall readability of the manuscript (which was in fact already quite readable), and helped to clarify a number of specific points.

In particular, my previous observations and requests of change are satisfactorily addressed in the new version. I thus have only two remarks about the latter, concerning i) the MGKBA and ii) the introduction of the proof of the Boltzmann’s H-theorem fo the e-ph systems.

With respect to i), there is now a reference in Sect. 5.1 to previous work [60].
I think it might be preferable to refer to [60] already in the introduction to the paper, for a broader historical context for MGKBA.

Concerning ii), I found the proof of the H-theorem quite interesting. My suggestion here is that it might be appropriate for readability to state more explicitly that the total entropy of the e-ph system is written as S_e+S_ph in the context of the Boltzmann equation formulation (in general, in the presence of interactions, eqs.104 too would take a more general form).

Aside from these two minor points (and the possible related small changes by the authors), I recommend that the paper is published in the present form.

Requested changes

Two minor optional changes, see above

  • validity: top
  • significance: top
  • originality: top
  • clarity: high
  • formatting: excellent
  • grammar: excellent

Author:  Gianluca Stefanucci  on 2024-02-27  [id 4327]

(in reply to Report 2 by Claudio Verdozzi on 2024-02-26)

We thank the reviewer for his/her suggestions. In the resubmitted version we have changed the manuscript accordingly.

---

## Round 2 · List of Changes

List of changes

1) We have realized that the full HSEX contribution can be included in Eq.(61). With the new definitions in Eqs.(62) and (63) the purely electronic SEPE becomes equivalent to the time-dependent HSEX approximation for vanishing scattering terms.

2) Brief discussions have been added following the suggestions of the Referees, see our reply to points 2, 4, 5, 8, 10, and 11 of Referee 1 and points 1, 6, 7, and 9 of Referee 2

3) Addition of Appendix C "H-theorem for coupled electron-phonon systems"

---

## Editorial Decision

published